# Regional Climate Change Impact on Coastal Tourism: A Case Study for the Black Sea Coast of Russia

Evgeniia A. Kostianaia [1] and Andrey G. Kostianoy [1,2,*]

1    P.P. Shirshov Institute of Oceanology, Russian Academy of Sciences, 36, Nakhimovsky Pr., 117997 Moscow, Russia; evgeniia.kostianaia@ocean.ru
2    Laboratory for Integrated Research of Water Resources, S.Yu. Witte Moscow University, 12, Build. 1, Second Kozhukhovsky Pr., 115432 Moscow, Russia
*    Correspondence: kostianoy@ocean.ru

**Abstract:** Regional climate change is one of the key factors that should be taken into account when planning the development of the coastal tourism, including investments and construction of tourism-related infrastructure. A case study for the Black Sea coast of Russia shows a series of potential negative hydrological, meteorological, and biological factors that accompany regional warming of the Black Sea Region, that can impede the development of coastal tourism and devalue billions of dollars in investments by the State, private companies, and individuals. We discuss such natural phenomena as air and sea warming, extreme weather events, coastal upwelling, heavy rains, river plumes, wind and waves, tornado, rip currents, sea-level rise, algal bloom, introduced species, and other features characteristic for the region that seriously impact coastal tourism today, and may intensify in the nearest future. Sporadic occurrence of extreme weather events, unpleasant and sometimes dangerous sea and atmosphere phenomena during the summer tourist season, and from year to year can be of critical psychological importance when choosing your next vacation and tourism destination. The research does not include anthropogenic factors, geopolitical, and socio-economic processes, and the COVID-19 pandemic that play an important role in the sustainable development of coastal tourism as well.

**Keywords:** Black Sea; coastal tourism; regional climate change; warming; wind; waves; sea level rise; upwelling; heavy rain; river plume; algal bloom; introduced species

## 1. Introduction

According to the UN Atlas of the Oceans [1], coastal tourism and recreation include the full range of tourism, leisure, and recreationally oriented activities that take place in the coastal zone and the offshore coastal waters such as recreational boating, cruises, swimming, recreational fishing, surfing, windsurfing, supping, snorkeling, diving, etc. These include the infrastructure supporting coastal development (e.g., hotels, resorts, restaurants, marinas, beaches, recreational fishing facilities, shops, roads, railways, airports, transportation). However, according to ECORYS [2], differentiation should be made between coastal and maritime tourism. For the purpose of this study, we shall use the definitions from [2], where coastal tourism would imply beach-based recreation activities (such as, for example, sunbathing, swimming, surfing, etc.), as well as non-beach related land-based tourism in the coastal area (this covers any recreation or tourism activities in the coastal area that require the sea to be in the proximity). Associated manufacturing industries and supplies also refer to this type. At the same time, maritime tourism refers to tourism that is mostly water-based than land-based, such as, for example, cruising, yachting, boating. This type also covers manufacturing of equipment, landside facilities, and necessary services.

The primary focus of this study is the impact of climate change on coastal tourism rather than maritime tourism. The main purpose of tourists coming to the Black Sea coast

of Russia is to sunbathe, swim, and snorkel, which falls under the coastal tourism category. Climate change effects seem to be able to impact coastal tourism in a more pronounced form as air and sea temperature warming, heavy precipitation and flooding, algal bloom, and introduction of alien species are expected to affect the possibility to lie on the beach, get into the water, and swim. These parameters seem to have a lesser impact on such maritime tourism activities as cruising, yachting, and boating.

Coastal tourism plays an important role in the socio-economic development of many countries. For some of the countries, it was an engine for economic development, other countries successfully used coastal tourism to overcome economic crises, and for others, it is the main source of their budget revenue. We have plenty of examples in Europe: the MENA region, Southeast Asia, the Caribbean, and other parts of the world. Every country which has an exit to a warm sea invests in the development of coastal tourism, because it brings an important income into state and regional budgets, provides employment and jobs to the local population. According to different sources, in 2006, the coastal tourism industry has contributed about 10% of the total global Gross Domestic Product (GDP) [1,3,4]. In fact, in 2019, prior to the COVID-19 pandemic, the travel and tourism industry accounted for 25% of all new jobs in the world, 10.6% of all jobs (334 million), and 10.4% of global GDP (USD 9.2 trillion). At the same time, international visitor spending totaled USD 1.7 trillion in 2019 (6.8% of total exports, 27.4% of global services exports) [3]. Most of these values belong to coastal tourism because this is the preferred destination for summer vacation.

The development of coastal tourism seriously impacts the coastal zone, terrestrial and marine environment, water and energy resources. Fresh and drinking water consumption, seawater, ground, air and noise pollution, plastic and microplastic pollution, untreated sewage, wastewater discharge into the sea, accumulation of waste and garbage, land degradation and land-use, coastal erosion, habitat and biodiversity loss, destruction of aesthetic value, and physical beauty of the coast are among the main threats caused by coastal tourism [4,5]. Uncontrolled tourism development in some cases can lead to environmental damage that can cost local populations and governments more than they would gain from the coastal tourism industry [4]. On the other hand, sustainable development of coastal tourism has the potential to create beneficial effects on the environment by raising awareness of environmental values, contributing to environmental protection and conservation, thus resulting in an increase in their economic importance [1]. The environment also can have both positive and negative impacts on the coastal tourism development, for example, via extreme weather events; physical, chemical, and biological processes in the sea; earthquakes and volcano eruption, and regional climate change which has been of great importance during the past 40 years, but usually ignored in strategies of coastal tourism development. All this leads to the necessity of sustainable coastal tourism management and development, which can be done, for example, in the framework of Integrated Coastal Zone Management [6], Marine Spatial Planning [7], and Maritime Clusters [8–11], which have been increasingly acknowledged as essential boosters for innovation and diversification of the Blue Economy [11].

In the Russian Federation, the Black Sea coast of Russia is the major resort area for Russian citizens, because the Arctic Seas are not taken into consideration, the Baltic Sea and the Sea of Japan are much colder, and the Sea of Azov and the Caspian Sea have a little resort area infrastructure in comparison with the Black Sea. Thus, the major coastal tourism flow is directed to the Krasnodar Krai (Region) and the Republic of Crimea (Figure 1). In 2019, 17.3 million tourists visited the Krasnodar Krai, 60% of them during the summer season, and 7.43 million tourists visited Crimea. These numbers include both foreign and domestic visitors [12].

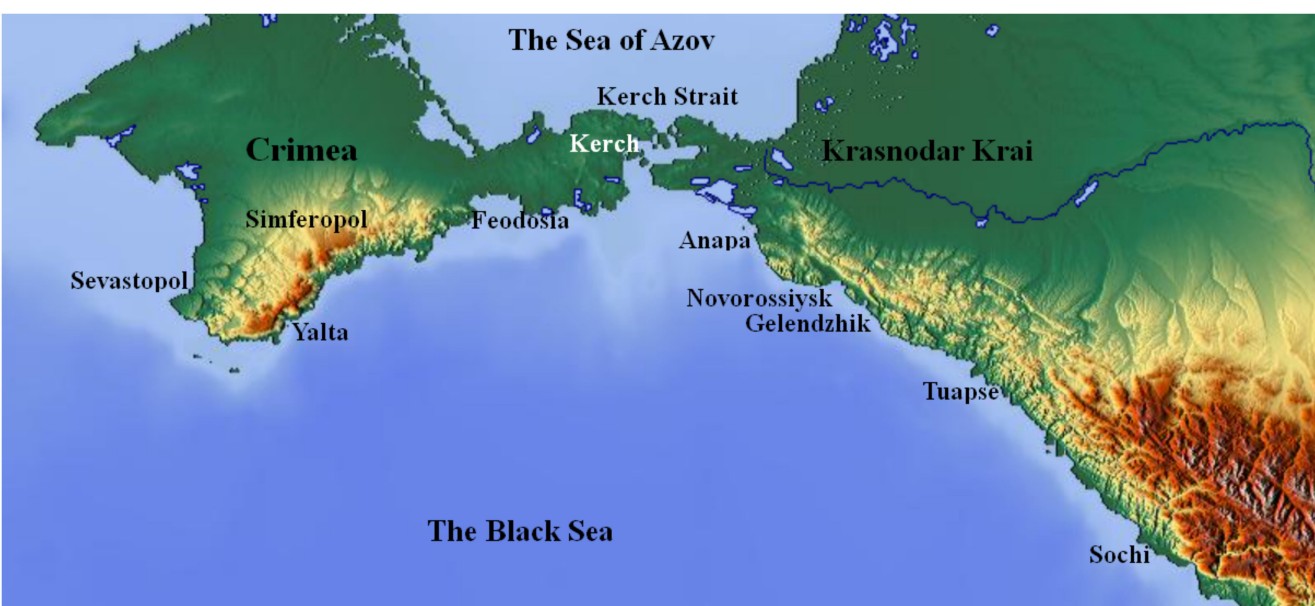

**Figure 1.** Geographical map of the Northeastern Black Sea based on Maps-For-Free (https://maps-for-free.com/#close, accessed on 30 August 2021). Blue line is the Kuban River. The Greater Caucasus stretches along the Black Sea coast.

The region under consideration is located between the Kerch Strait in the north and the state border with Abkhazia in the south, i.e., in the latitudinal band between 43°23′ and 45°30′ N (Figure 1). For comparison, this geographical band corresponds to the Northern Adriatic Sea between Split in Croatia and Venice in Italy, or the northern part of the Ligurian Sea between Cannes in France and Genoa in Italy. Both regions in the Mediterranean Sea are well known as the best places for coastal tourism and summer vacation in the Mediterranean due to a warm climate and warm sea. The same is true for the Russian coast of the Black Sea.

The climate in most of the territory of the Krasnodar Krai is moderately continental, on the Black Sea coast from Anapa to Tuapse—a semi-dry Mediterranean climate, south of Tuapse—humid subtropical. The high-altitude climatic zoning is characteristic of the mountains. In general, the region is characterized by hot summers and mild winters. The average January temperature on the Black Sea coast is 0 . . . +6 °C, in Sochi +6 °C. The average July temperature is +22 . . . +24 °C. Annual precipitation is from 400 to 600 mm in the flat part, and up to 3250 mm in the mountains [13].

The development of resorts on the Black Sea coast of Russia in the Crimea and the Krasnodar Krai began at the end of the 19th century. Today, this is a major resort and coastal tourism area for Russian citizens who spend their summer holidays on numerous beaches of the Black Sea. In December 2015, the Krasnodar Krai was named the most attractive tourist region of Russia in the "National Tourism Rating" (second and third places—St. Petersburg and Moscow). It includes resorts of federal significance Sochi, Gelendzhik, and Anapa, as well as numerous small villages and resorts of regional significance located between Anapa and Adler at the border with Abkhazia (from north to south—Vityazevo, Anapa, Dyurso, Yuzhnaya Ozereevka, Shirokaya Balka, Myskhako, Novorossiysk, Kabardinka, Gelendzhik, Divnomorskoe, Dzhankhot, Krinitsa, Betta, Arkhipo-Osipovka, Dzhubga, Lermontovo, Novomikhailovskiy, Olginka, Nebug, Agoy, Tuapse, Shepsi, Lazarevskiy, Vardane, Loo, Dagomys, Sochi, Matsesta, Khosta, Kudepsta, Adler) [14]. For instance, in Sochi, there are around 1000 certified hotels from 5-star (192) to mini-hotels (414) [15]. In Crimea, the most well-known resorts are located in (from west to east): Evpatoriya, Sevastopol, Balaklava, Simeiz, Alupka, Yalta, Gurzuf, Alushta, Sudak, Koktebel, Feodosiya, and Kerch [14].

In recent years, the most attention in the Krasnodar Krai has been paid to the development of a health resort complex, which makes it possible to increase the load of the health resorts in the off-season. The health resort sphere of the Krasnodar Krai includes more

than 200 organizations, their total capacity is about 100 thousand places. This is about 21% of the bed capacity of all Russian health resorts. On average, up to 1.5 million people annually rest and recover in health resorts, thus, the Krasnodar Krai occupies more than 25% of the Russian market of such services [12].

During the past 10 years, the Russian Government has invested significant funds in the development of tourism on the coasts of the Black Sea, including large infrastructure projects—reconstruction of airports in Sochi (Adler), Gelendzhik, Anapa, Krasnodar, and Simferopol; construction of the Crimean Bridge across the Kerch Strait and federal highway "Tavrida" from Kerch to Sevastopol; construction of Imeretinsky Resort and Olympic Park south of Sochi; building of new hotels and reconstruction of old ones; construction of the offshore gas pipeline "Dzhubga–Lazarevskoye–Sochi", and many others. All these projects have been performed with an Environmental Impact Assessment (EIA) which, according to the International Association for Impact Assessment (https://www.iaia.org, accessed on 30 August 2021), is "the process of identifying, predicting, evaluating and mitigating the biophysical, social, and other relevant effects of development proposals prior to major decisions being taken and commitments made".

Usually, the development of tourism and infrastructure on the Russian coasts of the Black Sea is carried out without assessment of the regional climate change impact in the nearest future and in the long-term perspective. To our knowledge, the same is typical for other resort areas in the world. This is not an obligatory procedure such as the EIA, but the ongoing climate change in the Black Sea Region forces such an analysis to be done to be sure that the investment will be as efficient as possible. The main feature of regional climate change in the Black Sea Region is the warming of the climate and seawater, which are regarded as evident and positive consequences of climate change for coastal tourism development. However, along with the warming of the region, climate change can lead to a series of negative processes (in relation to coastal tourism development) in the atmosphere and the sea which should be carefully studied before adopting long-term regional development programs.

Regional climate change in the Black Sea is accompanied by the intensification of extreme weather events. The Fourth [16] and Fifth [17] Climate Change Assessment Reports of the Intergovernmental Panel on Climate Change (IPCC) indicate that in the 21st century, climate change will be accompanied by an increase in the frequency, intensity, and duration of events with extremely high or low air temperatures, extreme precipitation or drought. All this may lead to floods, droughts, fires, shallowing of rivers, lakes, and water reservoirs, desertification, dust storms, algal bloom in the seas, and freshwater reservoirs. In turn, these phenomena in many cases may lead to chemical and biological pollution of water, land, and air, as well as to deterioration in the quality of life of the population, significant financial losses associated with damage to housing, businesses, roads, and railways, agriculture and forestry, coastal tourism, and in many cases even to human losses. The First [18] and Second [19] Roshydromet assessment reports on climate change and the consequences on the territory of the Russian Federation confirm these forecasts. Forecasts of independent groups of scientists have been repeatedly confirmed over the past 20 years—heavy rains, floods, droughts, and fires in various regions of the Russian Federation, including the coastal zone of the Krasnodar Krai, were yearly observed. In this regard, the analysis and forecasting of extreme weather events associated with regional climate change in the coastal zone of the Krasnodar Krai and Crimea are extremely important tasks, given the importance of coastal tourism for these regions.

Research interest towards the relationship between climate and weather, on the one hand, and tourism and recreation, on the other hand, became evident already in the 1950s [20]. A lot of attention has been given to evaluate predicted climate change by certain touristic regions, as consequences of such climatic changes will vary across regions and might mean different implications, increasing or decreasing the touristic flow. Such effects are manifold and have regional specifics. Detailed discussions can be found in publications from some of the major touristic coastal regions of the world, such as, for

example, published by: Becken [21] for Australia; Grimm et al. [22] for Brazil; Layne [23] for Caribbean SIDS; Becken [24] for Fiji; Friedrich et al. [25] for South Africa. The main discussed impacts for these regions are the same: higher temperatures, increased frequency, and intensity of extreme storms, increased precipitation, sea-level rise, sea temperature rise, change in ecosystems. Some other specific impacts are ocean acidification, coral bleaching, migration of species, the appearance of illnesses, change in the appearance of insects and organisms. Layne [23] and Lincoln [26] also point to the impact of climate change on human health and its implications for the tourist sector, citing such issues as extreme sunburn, dehydration, heatstroke, damage of public health facilities, water shortage during droughts, increase in skin diseases, cardio-respiratory conditions, and heat-related illnesses, spread of diseases through stagnant contaminated water. Some other indirect impacts are presented by Santos-Lacueva et al. [27]: increased energy consumption for cooling, increased water price due to scarcity, increased water consumption for recreation and comfort of tourists, artificialization of beaches due to sea-level rise, and the need for pest fumigation.

Klueva et al. [28] evaluated summer tourism for several regions in Russia, using the "tourism climate index"—TCI. For coastal tourism in the south of Russia, the authors came to the conclusion that climatic resources will increase towards the middle of the 21st century. Towards the end of the century, the TCI will lower, however, it will still remain within the range of comfortability for coastal tourism.

Despite the increasing significance of continuous research on climate change impacts on coastal tourism for the Black Sea coast of Russia, there are very few studies on this subject for this specific region. Extreme precipitation in summers, including this summer of 2021, has already led to significant disruption of transport services, evacuation of people, economic losses, and death tolls. Therefore, it is important to continue such research for this specific region to assess not only direct impacts on coastal tourism but also indirect consequences, including economic implications. A good example of such detailed research is a study by Arabadzhyan et al. [29], where the authors identified the following nine impact chains:

1.  Loss of tourist experience value in the destination due to changes in environmental attributes.

    1.1.  Loss of attractiveness of marine environments due to loss of species, increase in exotic invasive species or degradation of landscape.
    1.2.  Loss of attractiveness and comfort due to beach availability reduction.
    1.3.  Loss of attractiveness due to increased danger of forest fires in tourism areas.
    1.4.  Loss of attractiveness of land environments due to loss of species, increase in exotic invasive species or degradation of landscape.

2.  Loss of tourist experience value in the destination due to changes in human being comfort (or health).

    2.1.  Loss of comfort due to increase in thermal stress and heat waves.
    2.2.  Increase in health issues due to emergent diseases.

3.  Loss of tourist experience value in the destination due to the change in the quality of infrastructure and facilities.

    3.1.  Increase in damages to infrastructures and facilities (accommodation, promenades, water treatment system, etc.).
    3.2.  Decrease in available domestic water for the tourism industry.
    3.3.  Loss of attractiveness due to loss of cultural heritage (monuments, gastronomy, etc.).

According to this breakdown, our research concerns impact chains 1.1. (partially), 1.2., 2.1., and 3.1. Further directions would also need to concern other impact chains, for which it would be essential to cooperate and receive information from the tourist industry specialists, marine biologists, health workers, public utilities and cultural tourism representatives.

The aim of this paper is to review a series of potential negative hydrological, meteorological, and biological factors for the Black Sea coast of Russia that accompany regional warming of the Black Sea Region and can impede the development of coastal tourism and devalue billions of dollars of investments by the State, private companies, and individuals. We discuss such natural phenomena as air and sea warming, extreme weather events, coastal upwelling, heavy rains, river plumes, wind and waves, tornado, rip currents, sea level rise, algal bloom, introduced species, and other features characteristic for the region that seriously impacts coastal tourism today, and may intensify in the nearest future. In the paper, we do not investigate the impact of anthropogenic factors, geopolitical and socio-economic processes, and the COVID-19 pandemic that plays an important role in the sustainable development of coastal tourism as well.

## 2. Materials and Methods

The review of different meteorological, hydrological, and biological features and processes is made on the basis of scientific publications, our own research conducted for the Black Sea coast of Russia during the past 30 years, and our own experience as "tourists" coming to Novorossiysk for summer vacation every year.

The analyses and visualizations of data for the air temperature and atmospheric precipitation used in this study were produced with the Giovanni online data system v.4.32, developed and maintained by the NASA Goddard Earth Sciences Data and Information Services Center (GES DISC) [30]. As for the air temperature, we used the area-averaged 2 m air temperature (monthly mean) with a spatial resolution of $0.50 \times 0.625°$ for the region $43.4–45.2°$ N; $36.6–40.0°$ E, which covers the coastal zone of Russia from the Kerch Strait to the border with Abkhazia (Figure 1), and the time period from 1981 to 2020 (MERRA-2 Model M2IMNXASM_5.12.4) [31]. For the sea surface temperature, we used the area-averaged (monthly mean) skin water temperature produced by MERRA-2 M2TMNXOCN V5.12.4 Model with the same spatial resolution and for the same region [32]. For these parameters, we investigated interannual variability of monthly averaged data for the summer period only (June–August) as well as for May and September, separately.

For atmospheric precipitation (mm/month) we used an area-averaged "Merged satellite-gauge precipitation estimate final run" (GPM_3IMERGM v06) (monthly mean) model with a spatial resolution of $0.1 \times 0.1°$ for the same region and for the time period 2000–2020 [33]. For the wind speed (m/s), we used area-averaged monthly mean values of the wind speed from the MERRA-2 M2TMNXFLX v5.12.4 Model with a spatial resolution of $0.50 \times 0.625°$ for the same region and for the time period from 1981 to 2020 [34].

Satellite Synthetic Aperture Radar (SAR) image was downloaded from the ESA Copernicus Open Access Hub (https://scihub.copernicus.eu, accessed on 30 August 2021). For optical imagery we used NASA Worldview Snapshots (https://wvs.earthdata.nasa.gov, accessed on 30 August 2021).

## 3. Results

Regional climate change leads to a number of changes in the atmosphere and the sea, which potentially can be both positive and negative to socio-economic development. The most known consequence of climate change is warming, but this is not a single effect. Climate change is accompanied by changes in atmospheric circulation, the position of atmospheric fronts, trajectories of cyclones and anticyclones, atmospheric pressure, wind speed and direction, precipitation intensity and location, intensification of extreme weather events, and other processes and phenomena. In the sub-sections below, we try to describe the major natural features which are observed regularly during the past two decades at the Russian coast of the Black Sea, and we focus especially on the negative ones in relation to coastal tourism.

### 3.1. Warming of the Region and Extreme Events

Warming of the air and the sea is an evident consequence of global and regional climate change in the Black Sea Region. Ginzburg et al. [35] showed that the air temperature in the Black Sea Region is rising with a rate of +0.053 °C/year for 1980–2020, which is three times faster than for the 1935–2017 time period. The highest rate of change of +0.06 °C/year is located along the coastal zone of the northeastern Black Sea. Since the late 1990s, the maximum monthly mean values of the near-surface air temperature during summer on average increased considerably, with an extreme value of 27.2 °C in 2010, when a blocking anticyclone stood over the central part of European Russia for 55 days since the end of June [36]. The above-mentioned analysis, as well as previous publications on regional climate change in the Black Sea Region mentioned in the review [37], showed air temperature trends for all 12 months of the year during a certain time period. In the present study, we are interested in what is happening during summertime, which is a tourist season. This is why we calculated the interannual variability of the air temperature for the summer season (June, July, August) for a 40 years-long time period from 1981–2020 (Figure 2).

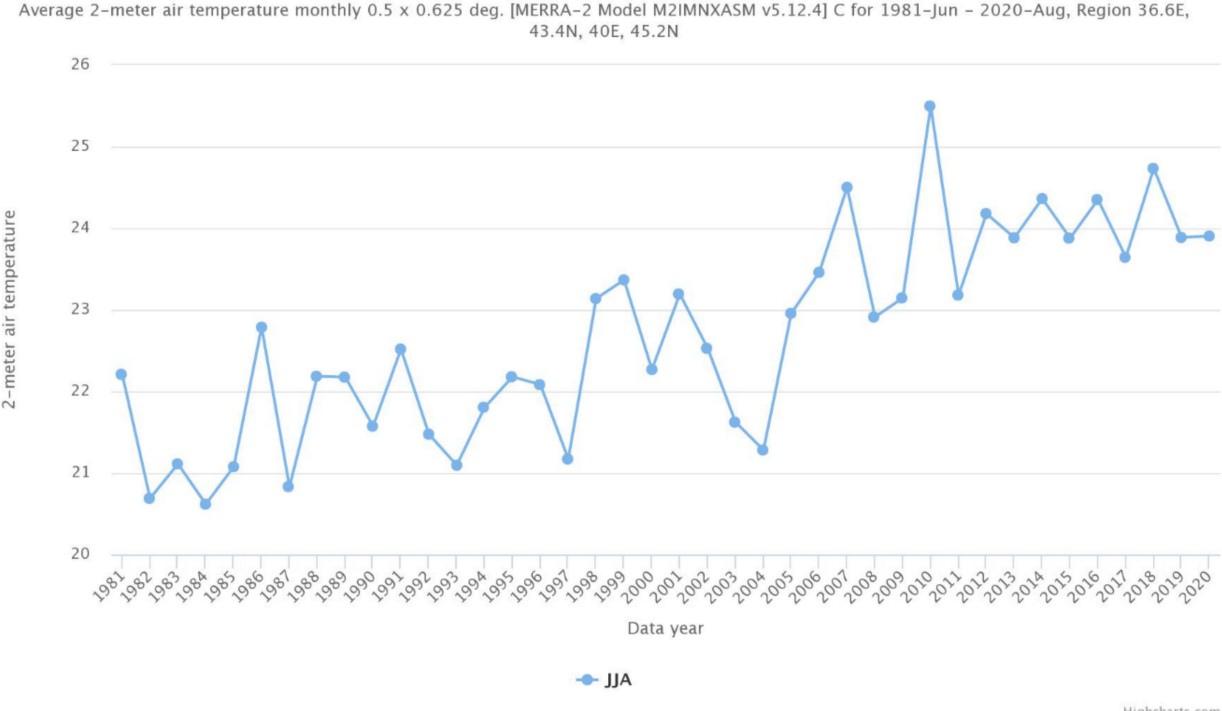

**Figure 2.** Interannual variability of the near-surface air temperature over the northeastern Black Sea during summer from 1981–2020 based on the MERRA-2 M2IMNXASM_5.12.4 Model.

Figure 2 confirms that, on average, the air temperature during summer has been progressively warming over the past 40 years from around 21 °C in the early 1980s to around 24 °C in the 2010s. The transition between the period of a slight increase in the air temperature to a more stable period occurred between 2004 (21.3 °C) and 2012 (24.2 °C). The hottest summer, 25.5 °C, was registered in 2010 due to a long-standing blocking of an anticyclone over the central part of European Russia. Year-to-year variability of the air temperature varied from 0.5 to 2.5 °C from 1981 to 2011, but since 2012 the air temperature variability has become more stable and has not exceeded 1 °C.

These data show that the air temperature, on average, gets warmer. The air temperature is becoming more comfortable for coastal tourism for a longer period of time, and the tourist season slowly expands for May and September. The same analysis for

May (Figure 3) showed that the air temperature rose from around 15 °C at the end of the 20th century to 16 °C in the first decade of the 21st century, and 17 °C in the 2010s. First, the average air temperature increase in May is lower than the same parameter for the average value for the summertime. Second, 17 °C does not seem to be comfortable for opening the tourist season. Third, the weather in May is very unstable, because from year to year the air temperature has varied from 15 to 19 °C over the past 10 years. Thus, it is evident that during the coming decades, May will not be a month from which the tourist season will start. At the same time, September shows to be a better candidate for expansion of the summer season because its average air temperature is 3–4 °C higher than in May, and it is already comparable with June (Figure 4). Since 2005, its average temperature almost yearly has been over 20 °C, which can be regarded as a psychological mark for summertime, and in 2015, 2017, and 2020 it was over 22 °C.

Global and regional warming is accompanied by extreme weather events such as heat and cold waves, draughts and frosts, heavy rains and snowfalls, etc. This effect is also observed in the eastern part of the Black Sea which was especially investigated for the period 1950–2015 based on daily air temperature data [38]. Kostianoy et al. [38] showed an increase in the amplitude of air temperature extremes with positive anomalies from 3.6 °C to 3.9 °C for phenomena exceeding one standard deviation (1 SD), and from 5.5 °C to 6 °C for phenomena exceeding 2 SD. At the same time, the amplitude of extreme events with negative anomalies remained practically unchanged: 3.9 °C and 7.2 °C, respectively. The number of extreme events with positive anomalies exceeding 1 SD increased from 10–14 to 28–32 events per year, and exceeding 2 SD—from 1–2 to 12–14 events per year. At the same time, the number of extreme events with negative anomalies exceeding 1 SD decreased from 22–24 to 8–10 events per year, and exceeding 2 SD—from 5–6 to 2–3 events per year. The average duration of extreme events with positive anomalies exceeding 1 SD increased from 2.5 to 3.5 days, but with negative anomalies, practically remained unchanged—3 days. The duration of extreme events with anomalies exceeding 2 SD increased from 1 to 2 days for positive anomalies events and remained the same (2 days) for negative anomalies [38].

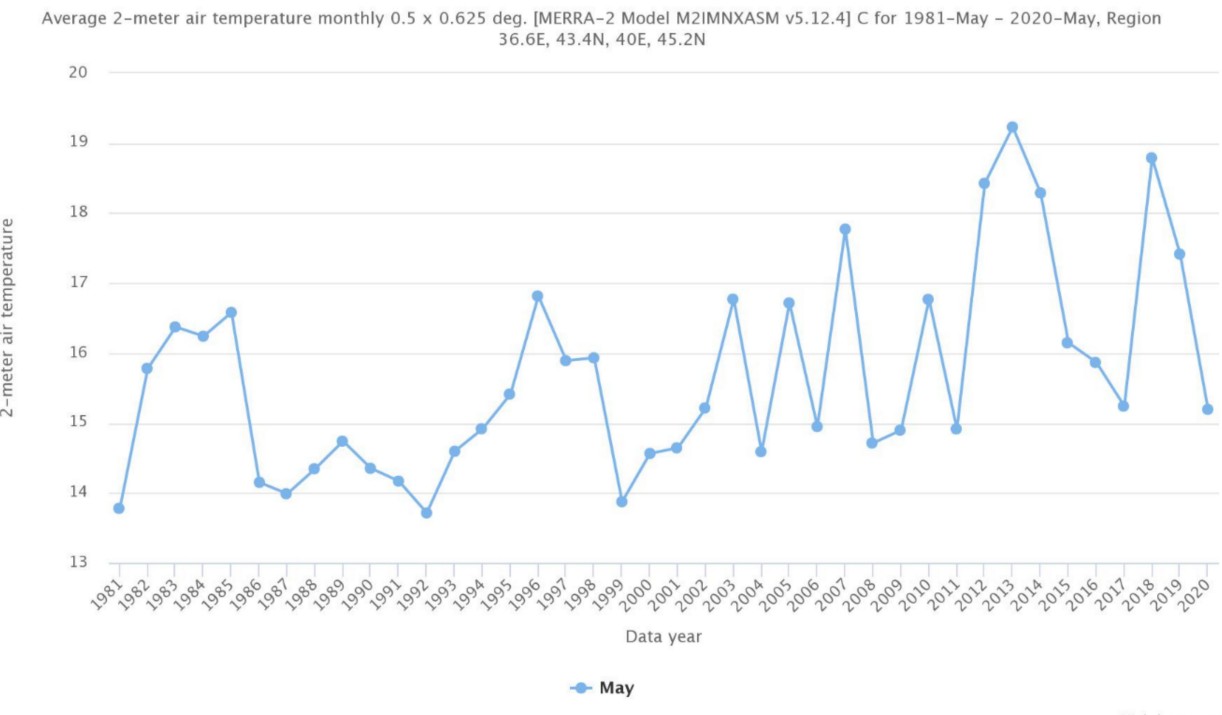

**Figure 3.** Interannual variability of the near-surface air temperature over the northeastern Black Sea in May from 1981–2020 based on the MERRA-2 M2IMNXASM_5.12.4 Model.

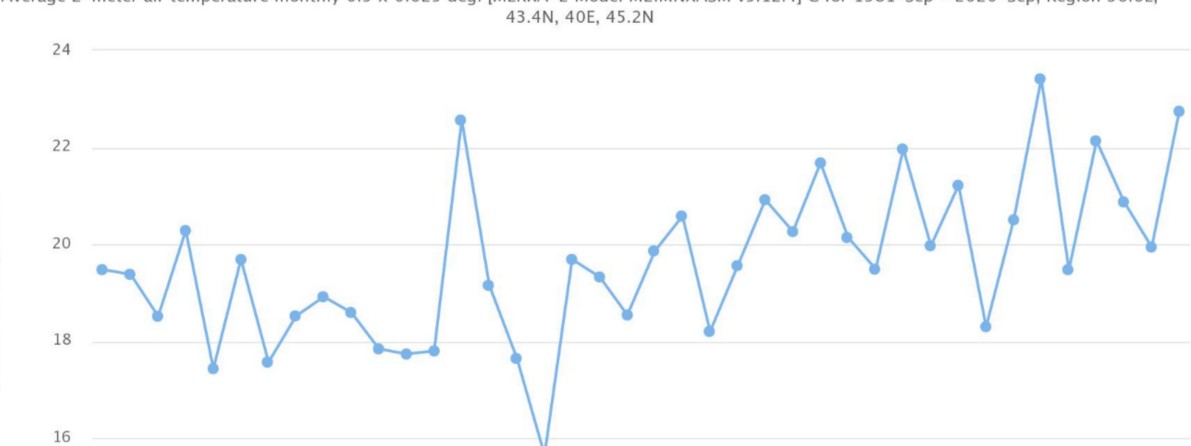

**Figure 4.** Interannual variability of the near-surface air temperature over the northeastern Black Sea in September from 1981–2020 based on the MERRA-2 M2IMNXASM_5.12.4 Model.

It means that, on average, heat waves in the Eastern Black Sea became a bit stronger, their frequency doubled for anomalous events exceeding 1 SD and reached 28–32 events per year, while strong events exceeding 2 SD increased 10 times from 1–2 to 12–14 per year, and the duration of both types of extreme events increased as well.

### 3.2. Sea Warming

The sea surface temperature (SST) is a second important factor for coastal tourism. The SST in the Black Sea, like the air temperature, has significantly increased over the past 40 years. Since the end of the 1990s, the maximum monthly average summer SST values have increased. Summer highs in most cases exceeded 25 °C with an extreme of 26.94 °C in August 2010. The maximum/minimum summer SST values correspond to approximately the same years as for the air temperature. The SST linear trend for the whole Black Sea for 1982–2020 is equal to + 0.052 °C/year, which is identical to the air temperature rate of change. The SST trends are unevenly distributed over the Black Sea area: the highest values (+0.058–0.060 °C/year) are observed along the northeastern coast of the sea (Russian coast), the minimum (less than + 0.044 °C/year)—in the center of the Western Black Sea (in the area of the western cyclonic gyre) and at the northwestern shelf of the sea [35]. Roughly the same picture of the distribution of SST trends over the sea area for the period 1981–2015 was presented in [39]. In the northeastern part of the sea from 1983–2015, the SST trends were recorded within + 0.075–0.084 °C/year [40].

In the present study, we are interested in the SST variability during summertime, this is why we calculated the interannual variability of SST for the summer season (June, July, August) only for a 40-years-long time period (1981–2020) and for the northeastern part of the Black Sea (Figure 5). We found that, on average, in the 1980s, the SST was around 22 °C, in the 1990s—23 °C, in the 2000s there was a sharp rise of the SST from 23 °C to 26 °C in 2010, and in the 2010s the SST stabilized around 25 °C. We have to note that features of interannual variability of the SST are similar to those characteristics of the air temperature (see Figure 2).

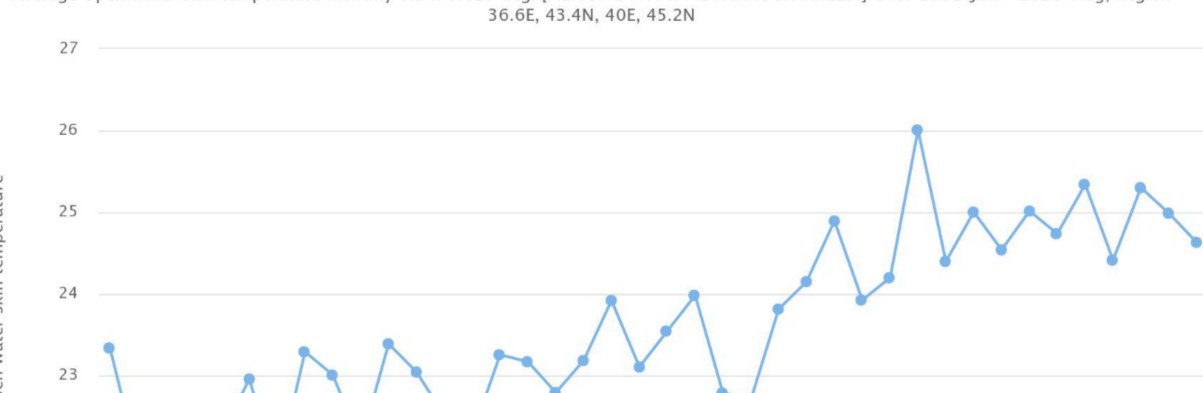

**Figure 5.** Interannual variability of the sea surface temperature in the northeastern Black Sea during summer from 1981–2020 based on the MERRA-2 M2TMNXOCN V5.12.4 Model.

We also repeated the same analysis for May and September (Figures 6 and 7) to understand in what direction the summer hydrological season is expanding. May shows a slowly warming, but only in 2012, 2013. and 2018 the SST was over 18 °C, and in the past decade, the SST varied from 14.76 °C in 2011 to 18.76 °C in 2018. Thus, the SST in May became much more unstable in comparison with the previous three decades and the SST remains too cold to open the swimming season (Figure 6). Similar to the case with the air temperature, September is already becoming a fully-fledged tourist month suitable for comfortable swimming because the SST during the past two decades has varied between 21.75 °C in 2003 and 24.87 °C in 2020 with a steady tendency to increase (Figure 7). September and October of 2020 were exceptionally warm, and it was possible to swim in the coastal zone of Russia till the end of October.

We also calculated the relationship between the air temperature and SST for the region under investigation and the same time period in the form of the scatter plot shown in Figure 8. It shows a relationship between the monthly mean 2m air temperature and the monthly mean SST. It is interesting that in the middle of the graph, i.e., between 5 and 20 °C of the air temperature, there is a hysteresis curve where there are different values of SST depending on the direction of change of the air temperature during a change of seasons. This is an interesting and evident result which is explained by the fact that in spring the air temperature is rising much faster than warming of the sea surface, and in autumn with a decrease in the air temperature, SST remains high for a longer period of time. Thus, at the same values of the air temperature, SST is higher in autumn than in spring.

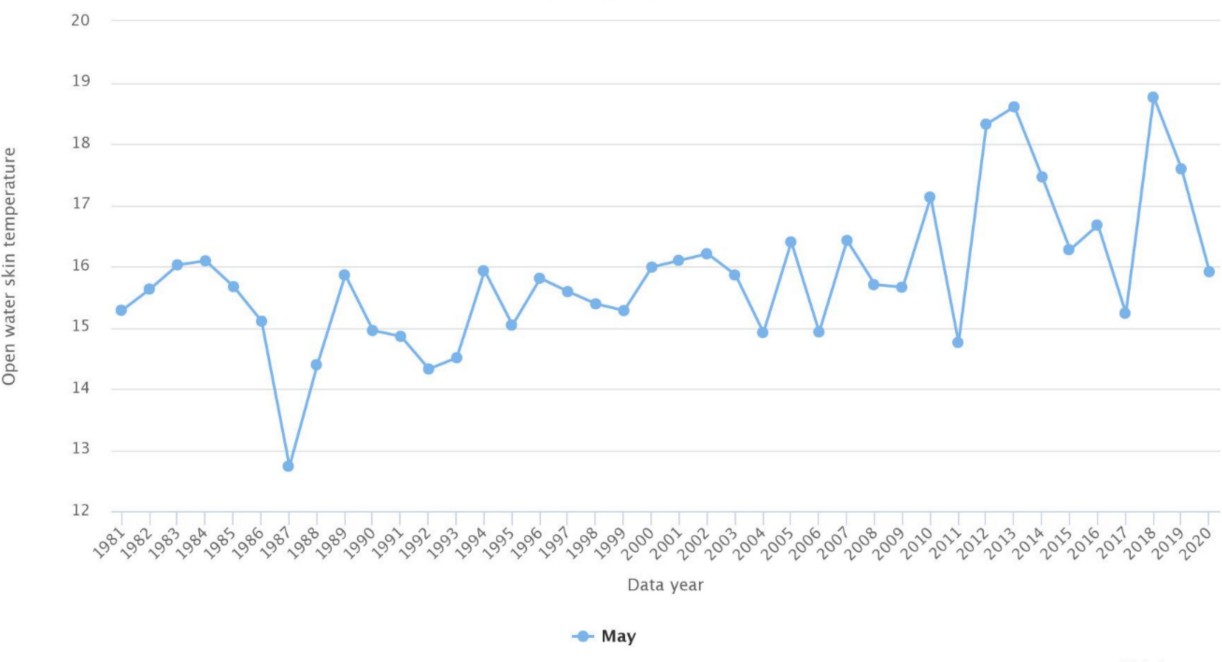

**Figure 6.** Interannual variability of the sea surface temperature in the northeastern Black Sea in May from 1981–2020 based on the MERRA-2 M2TMNXOCN V5.12.4 Model.

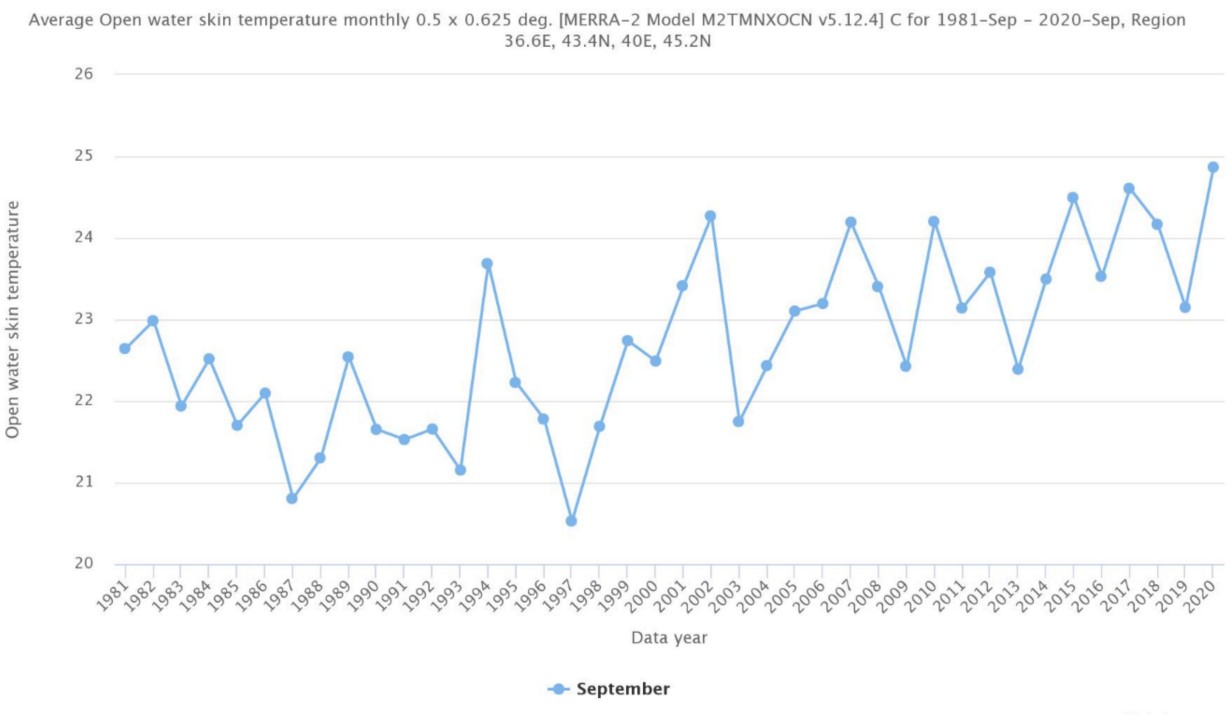

**Figure 7.** Interannual variability of the sea surface temperature in the northeastern Black Sea in September from 1981–2020 based on the MERRA-2 M2TMNXOCN V5.12.4 Model.

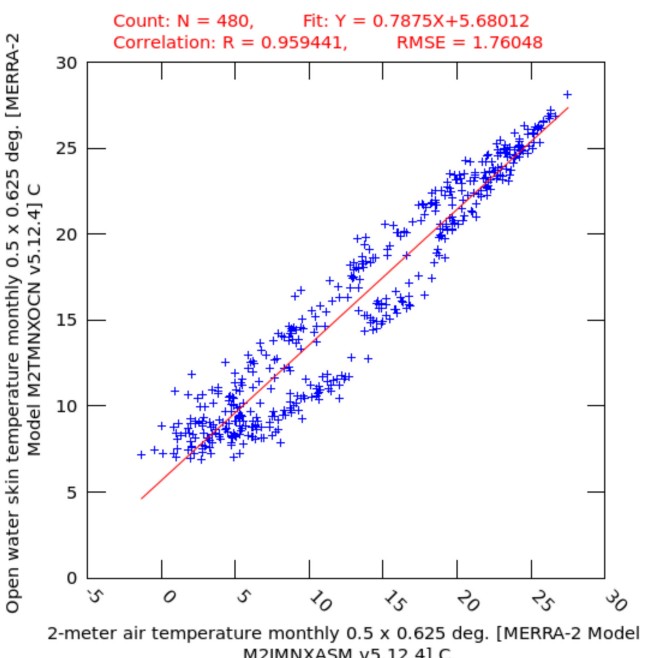

**Figure 8.** Relationship between the monthly mean 2−meter air temperature and the monthly mean SST for 1981–2020.

*3.3. Heavy Rains and River Plumes*

Heavy rains have become a serious problem for the coastal zone of the Krasnodar Krai of the Russian Federation and the Black Sea coast of Turkey. They lead to a significant increase in the water level in rivers, flooding of villages and even large cities, damage to urban and tourist infrastructure, roads and railways, bridges, beaches, and the washout of garbage and sewage into rivers. This leads to environmental problems and even human casualties.

For example, on 8 August 2002, the city of Novorossiysk and the nearby resort area Shirokaya Balka were flooded as a result of heavy rain, where, in less than a day there fell an equivalent of six-month precipitation (Figure 9). The tourist infrastructure was destroyed in Shirokaya Balka. According to official figures, almost 20,000 people were affected. More than 100 people died in the flooded area [41]. On 6–7 July 2012, according to the Hydrometeorological Center of Russia, the equivalent of five-month precipitation (275 mm) fell overnight in Krymsk, Novorossiysk, and Gelendzhik. One hundred and seventy-one people died in the flood, which damaged nearly 13,000 homes and affected nearly 30,000 more. The flooding was part of the aftermath of a severe storm that hit the Krasnodar Krai, which resulted in precipitation of the equivalent of almost six months in two days, and it was the largest in the last 70 years. Heavy rains caused dangerous flooding on rivers, including a catastrophic flood on the Adagum River near the city of Krymsk. The total damage is estimated at RUB 20 billion (USD 0.3 billion). Such torrential rains occur annually, for example: 7–8 September 2018, when torrential rains hit the entire coastal zone from Adler to the Kerch Strait; 24–25 October 2018, when torrential rains destroyed roads, railways, and bridges in the Tuapse region; 24 June 2019, when torrential rains caused landslides and damaged roads and tourist infrastructure in the mountain resort area of Krasnaya Polyana near Adler; 29 June and 16 July 2019, when torrential rains hit the city of Sochi [42].

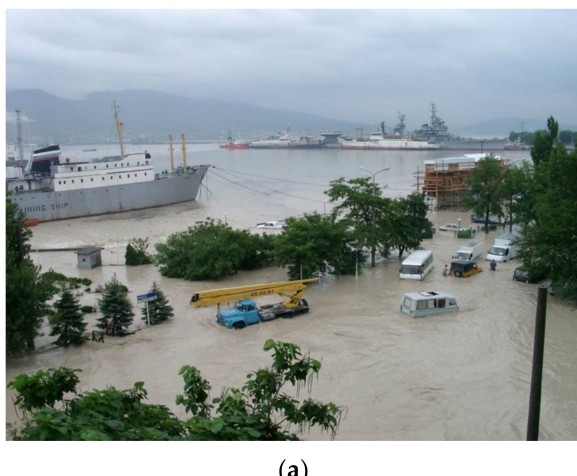
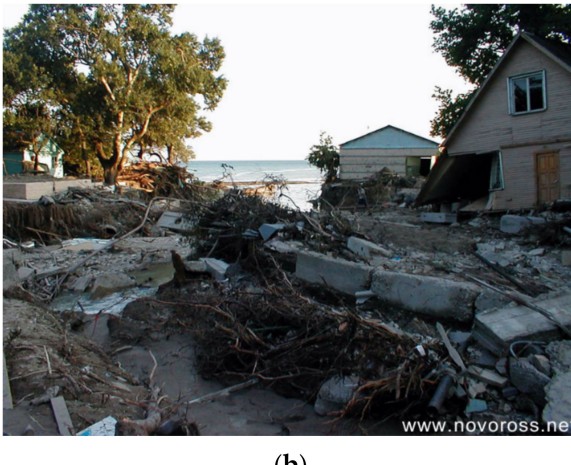

(**a**)                                                                                      (**b**)

**Figure 9.** (**a**) Flooded streets and embankment in the center of Novorossiysk on 8 August 2002; (**b**) damaged tourist infrastructure in Shirokaya Balka on 8 August 2002.

The summer of 2020 was rather dry, however the summer of 2021 again witnessed catastrophic rains and flooding. On 17 and 18 June 2021, heavy rains hit the coast of the Krasnodar Krai and of Crimea. In Yalta, the monthly norm of precipitation fell on the first night, in Kerch—2 norms of precipitation. Houses, streets, roads, the federal highway "Tavrida" turned out to be flooded. The hailstorm broke through the windows of cars and roofs of houses in some districts of the Krasnodar Krai. Following this, on June 18 in Yalta, another three monthly precipitation rates fell. For three weeks, the forces of the Ministry of the Russian Federation for Civil Defence, Emergencies, and Elimination of Consequences of Natural Disasters were engaged in dismantling debris, clearing streets, pumping out water, restoring housing, flooded and damaged infrastructure, and cleaning streets. The summer vacation was completely ruined, and the beaches were closed for three weeks. People returned tickets, canceled hotel reservations, and returned home.

On 26 June 2021, heavy rains in Gelendzhik and Anapa (50 mm for one hour) flooded the centers of cities, as well as damaged embankments and beaches. On 4 and 5 July 2021, the cities of Adler, Sochi, Lazarevskoye, Tuapse, Lermontovo, Dzhubga, Novorossiysk, and Anapa were flooded again as a result of heavy rains and the storm surge that occurred along the whole Black Sea coast of Russia from Adler to Kerch (Figure 1). In the Khosta Region of Sochi, one monthly norm of precipitation fell during the day, in the Bakhchisaray Region of the Crimea—2–3 norms of precipitation. More than 300 houses, roads, streets, embankments, beaches, gas and water supply, and hundreds of cars were flooded and broken. The only road leading from Dzhubga to Sochi along the entire Black Sea coast was closed for several days due to landslides and destructions by mudflows in several places. Eight people died, and several dozens were injured.

More heavy rains followed on 23 July 2021 (floods in Sochi and Khosta), 8 August 2021 (floods in Sochi, Tuapse, Novorossiysk, and Kerch), 10 August 2021 (flood in Novorossiysk, 4.5 mm during 1 h, Anapa, Vityazevo), 12–13 August 2021 (floods in Novorossiysk, Anapa, Kerch—2–4 monthly precipitation norm or 40–200 mm during 1 day), 15–16 August 2021 (floods in Gelendzhik, Novorossiysk (100 mm), Anapa (219 mm), Kerch (176 mm), 2000 houses were flooded. Water in Anapa was present till 20 August, 500 houses were still flooded, as well as a dozen power electricity stations which were switched off); 18–19 August 2021 (floods in Anapa—monthly norm).

We calculated monthly average precipitation over the northeastern Black Sea for June–August from 2000 to 2020 and found that since 2007 there was a steady rise of precipitation from 39.2 mm/month in 2007 to 86.8 in 2014 and 88.3 mm/month in 2017 (Figure 10). A peak in 2002 was partially related to the catastrophic rains on 21–22 June and 8 August in the area of Novorossiysk. A relative minimum of 2020 (58.3 mm/month) was related to the drought which was observed from summer 2019 till December 2020.

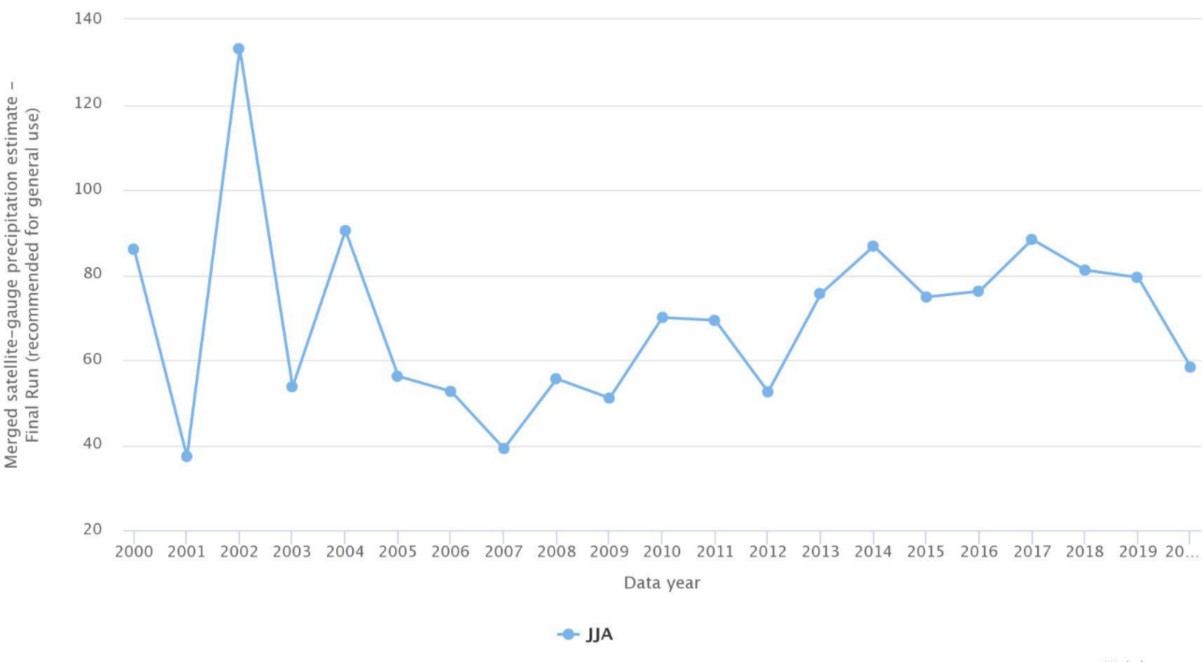

**Figure 10.** Interannual variability of atmospheric precipitation (mm/month) in the northeastern Black Sea in summer from 2000–2020 based on the GPM_3IMERGM v06 Model.

Looking at Figure 10, we cannot say that there are changes in heavy rains because: (1) the graph shows average values of total precipitation (mm/month) for three summer months; (2) the shown parameter is average monthly precipitation measured in mm/month but not in mm/hr which should be taken from records at meteorological stations; (3) the values are averaged over a large area. Zolina et al. [43,44], Zolina and Bulygina [45], based on records from meteorological stations, showed that in Europe, including the Black Sea coast of Russia, the structure of precipitation has changed to more abundant rainfalls. During 1950–2008, the occurrence of the association of heavy rainfall with longer WP (wet periods defined as consecutive days with significant precipitation > 1 mm/day) was increasing by 3–4% per decade. The occurrence of intense precipitation associated with longer WPs increased from 40% in the 1950s–1960s to 55% in the 1990s–2000s. Heavy rainfalls associated with longer WPs intensified over Europe with upward trends of 2–3% per decade in Western Europe and >5% per decade in European Russia, implying actual changes from 4 to 9 mm/day over the 60-year period. The changing character of WPs (short rain events have been regrouped into prolonged wet spells) may significantly alter the frequency and strength of floods. The highest values of the thresholds for absolute extreme precipitation (more than 26 mm/day) are observed in the North Caucasian Federal District (including the Caucasian Black Sea coast), which is characterized by a complex orography and the presence of mountain systems [45]. They found that for all seasons of the year, practically throughout the entire territory of Russia, there is a steady tendency towards an increase in the values of extreme precipitation up to 8% per decade (values for 1966–2012), which corresponds to an increase in the absolute values of extreme precipitation by about 4 mm/day over the last 50 years in winter and about 7 mm/day in summer.

Besides damage to coastal infrastructure, numerous rains lead to the cooling of seawater in the coastal zone and may delay the establishment of the comfortable conditions for swimming in the tourist season. Such a case was observed in June 2021, when seawater was colder than usual by about 5 °C, while the air temperature was quite typical for this month. The other problem related to heavy rains and river plumes concerns seawater

quality, which is very dirty, with a very high concentration of suspended matter and all kinds of pollutants gathered on land. Usually, after heavy rains, people do not swim in the sea for 1–2 days, in case of repeated heavy rains, and beaches can be officially closed for swimming for a week or more. This was the case in Yalta in the second half of June 2021.

River runoff forms river plumes in the coastal zone of the sea (Figure 11), which, depending on the speed and direction of the wind, the velocity, and direction of the coastal current, the magnitude of the runoff, and the difference in density with seawater, can have different forms and spread in different directions from the river mouth, as well as participate in the mesoscale and sub-mesoscale circulation of coastal waters [42,46–50]. River plumes significantly affect the quality of seawater and the sanitary and epidemiological situation on the beaches of the resort area of the Krasnodar Krai. The situation is exacerbated by the fact that sewerage systems are in poor condition in cities, or there are no such systems at all in most small villages along rivers and coastal areas. After heavy rains, wastewater very often reaches the sea and poses a serious threat to human health.

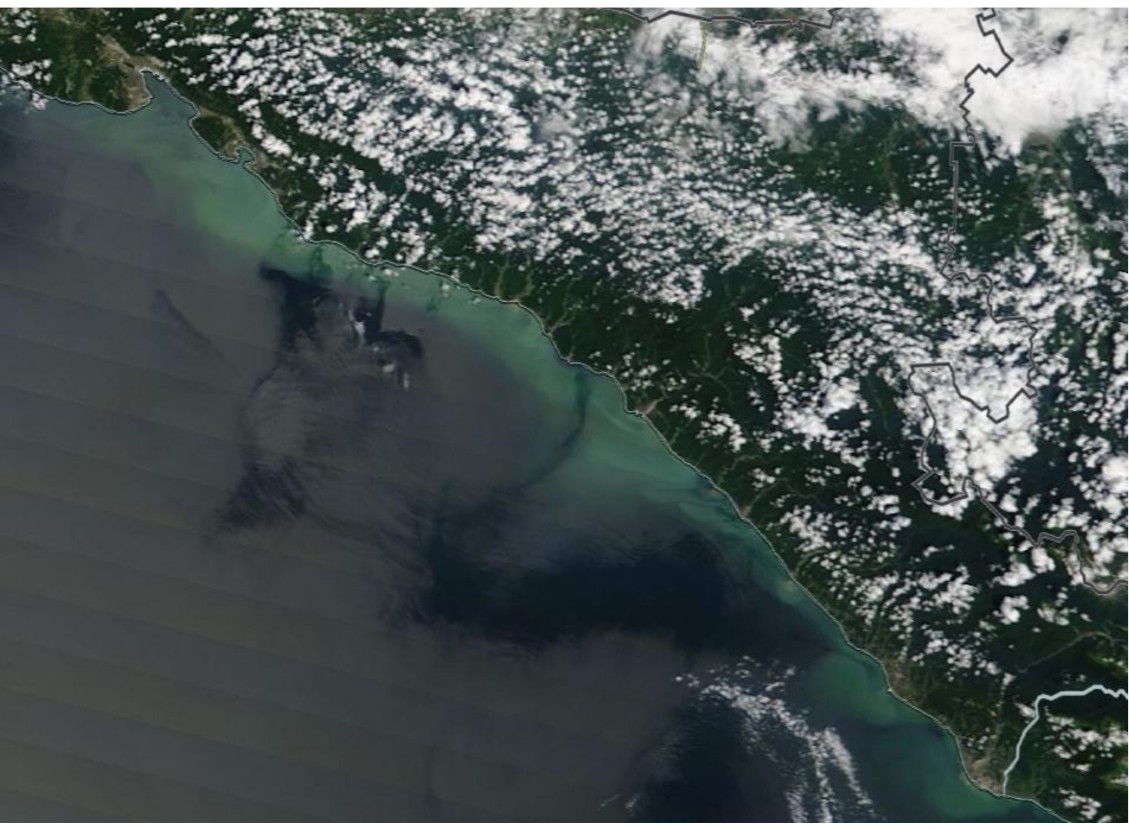

**Figure 11.** Joint river plumes (light green colors) along the Russian coast of the Black Sea from Novorossiysk to Adler and the border with Abkhazia on 7 July 2021 (Terra MODIS True Color Corrected Reflectance, https://wvs.earthdata.nasa.gov/api/v1/snapshot?REQUEST=GetSnapshot&LAYERS=MODIS_Terra_CorrectedReflectance_TrueColor,Coastlines_15m, Reference_Features_15m&CRS=EPSG:4326&TIME=2021-07-07&WRAP=DAY,X,X&BBOX=43.335571,37.41211,44.84436, 40.327148&FORMAT=image/jpeg&WIDTH=1327&HEIGHT=687&AUTOSCALE=TRUE&ts=1626260065393, accessed on 14 July 2021).

Figure 11 shows a satellite view of the coastal zone of the northeastern Black Sea on 7 July 2021 after heavy rains occurred along the whole coast on 4–5 July. Light green colors show numerous river plumes generated at the mouths of small rivers located along the coast. Heavy rains generate huge river runoff which results in merging of separate river plumes into a single turbid water mass which is characterized by a high concentration of suspended matter and all kinds of pollution. River plumes propagate northwestward

along the coast with a general current. Such a situation persists for several days until the flood flow of rivers stops and coastal waters mix with those around them.

### 3.4. Wind and Waves

Regional climate change-induced coastal processes such as sea-level rise, coastal flooding and erosion, and storm surge are the main reasons for coastal infrastructure damage and vulnerability [51–55].

Wind speed is one of the main climatic parameters. The wind has a significant effect on the exchange of momentum, heat, moisture, and trace particles between the atmosphere and the underlying ocean and land. It causes waves in the oceans and seas, storm surges, and has a huge impact on sea ice. It plays a key role in ocean circulation, which is responsible for the global transport of significant amounts of heat and carbon. Wind speed is a sensitive indicator of the state of the global climate system. Surface wind data are directly applied to industries such as transportation, construction, power generation, agriculture, human health, maritime safety, and emergency management [56].

In general, the Black Sea coast of Russia is not a windy region in summer. Strong winds and stormy conditions are usually observed in late autumn and winter. The most known natural phenomenon is Novorossiysk bora, a strong wind blowing from the northeast, with maximum occurrence and force in winter. Variability of the main directions of winds above the Black Sea is determined by the seasonal variability of the distribution of atmospheric pressure. In winter, north-easterly, northerly, and north-westerly winds prevail over the sea, and easterly winds prevail in its southeastern part. In summer, north-westerly, westerly, and south-westerly winds blow most often. In the cold half-year in the western part of the sea, the average wind speed reaches 7–8 m/s, and in the coastal zone—less than 7 m/s. In the southeastern part of the sea, the wind speed is 5–6 m/s, in the northeastern part—6–7 m/s. In the warm season, the average wind speeds are 1–1.5 m/s less, but they also increase from east to west. The lowest wind speed throughout the year is observed near the southern coast of Crimea and in the southeastern part of the sea [14].

Strengthening of the wind over the sea is most often due to the passage of atmospheric cyclones. The strongest wind in the Black Sea is Novorossiysk bora (or Nord-Ost, northeasterly wind), which is observed several times annually in autumn and winter (40–50 days) (Figure 12). The speed of the northeastern wind reaches 40 m/s, and with gusts—up to 80 m/s. Bora is often accompanied by a temperature drop of 10–20 °C. Its duration ranges from 1–3 days to a week and can affect from several dozens to 150 km offshore. The appearance of bora is limited to the area from Anapa to Tuapse, where the height of the mountains is not a barrier for such winds (Figure 1) [14].

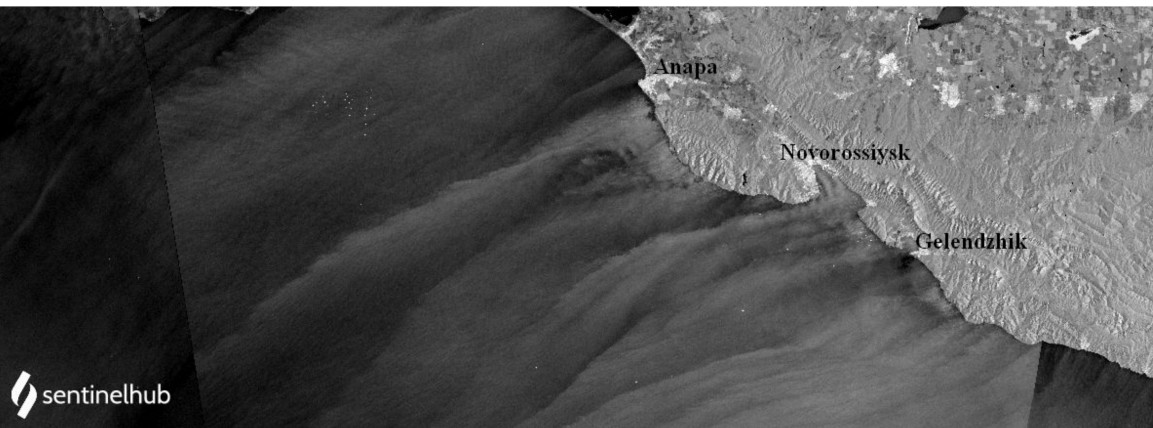

**Figure 12.** The impact of Novorossiysk bora on the coastal zone of the northeastern part of the Black Sea as revealed by Sentinel-1 SAR image on 5 August 2020. Grey bands propagating from the shoreline are a rough sea generated by the northeasterly wind. Ground length between Novorossiysk and Gelendzhik is 30 km.

During the past decades, the northeasterly wind starts to appear more often during the summertime when it is hot and dry. Its direction is almost perpendicular to the shore in the Krasnodar Krai, thus, even in summer, it is dangerous for swimmers as it generates strong offshore currents directly from the shoreline (Figure 12). In those places where the beaches are located under the cliffs, swimming is even more dangerous, since the cliffs shade the coastal strip 50–100 m wide from the wind, and then the waves and currents begin to accelerate, directed into the open sea, which swimmers cannot resist. Annually, several people are carried out to sea on inflatable mattresses at distances from 1 to 10 km, who have to be rescued on motorboats and even helicopters.

Kostianaia et al. [56] analyzed the seasonal and interannual variability of the wind speed in the eastern part of the Black Sea for the period from 1980 to 2013. In addition, a detailed analysis of extreme wind phenomena for the coastal regions of the Krasnodar Krai and Abkhazia was carried out, interannual variability of the amplitude, frequency, and duration of extreme wind events were studied. Changes in the monthly mean values of the wind speed module, calculated for the coastal region of the Krasnodar Krai (43.5–45° N; 37–40° E), show zero change in the average monthly value of the wind speed module. The average value for this period was approximately 2.3 m/s. The maximum monthly mean value of the wind speed module was observed in 1994 and amounted to approximately 9.2 m/s. On the coast of the Krasnodar Krai, there were a great number of maximum monthly mean values of the wind speed module: there were six cases when the monthly average wind speed was more than 6.5 m/s [56].

The analysis of extreme wind events has shown that there was an increase in the amplitude of extreme events with positive anomalies of the wind speed module from 4.35 m/s to 4.7 m/s for phenomena exceeding 1 SD and for phenomena exceeding 2 SD, there was a slight decrease in the amplitude of extreme events from 7 m/s to 6.8 m/s. The number of extreme events with positive anomalies exceeding 1 SD decreased from 33 to 32 events per year, and those exceeding 2 SD increased from 8 to 12 events per year. The average duration of extreme events with positive anomalies exceeding 1 SD increased from 1.65 to 1.75 days, and those exceeding 2 SD remained the same—1.38 days. In 1995, the maximum number of positive anomalies (44 events) in the wind speed module exceeding 1 SD was recorded [56].

We calculated interannual variability of the monthly averaged wind speed for June–August for 1981–2020 and found great variability from year to year from 5.6 in 1994 to 4.4 m/s in 1997. Wind conditions became more stable in the past decade with variations between 4.75 and 5.1 m/s (Figure 13).

A tornado is a very intense rotating column of air that contacts the cumulus clouds and the underlying surface. In the coastal zones of the seas and oceans, tornadoes are called waterspouts. They rotate with a velocity up to 180 km/h, have a diameter of several dozen meters, and can travel several kilometers before dissipating, sometimes coming to the shore [57]. This is a typical meteorological feature for tropical and subtropical waters and coastal zones of the USA, South America, Southeastern Europe, Southern Africa, India and Bangladesh, Australia, and New Zealand. Several decades ago, a tornado was a very exotic event in the Black Sea. Today, with progressive warming of the sea, in the northeastern part of the Black Sea, there are several cases of tornadoes every year with a maximum occurrence between June and September. Their coming ashore is sometimes accompanied by catastrophic consequences [57]. For example, in June 1991, residents of the city of Tuapse were seriously affected, and in August 2002, the city of Novorossiysk and its suburbs (Shirokaya Balka) were hit with a tornado and heavy rains (Figure 9). Houses, mini-hotels, power lines, roads, and railways were damaged, and several villages around were destroyed. Dozens of people died and went missing. More often, tornadoes occur along the beaches of Big Sochi where the sea surface temperature is the highest along the Black Sea coast of Russia. However, in recent years, waterspouts were registered in Gelendzhik, Novorossiysk, and even in Rostov-on-Don (2017 and 2019) located more northward, in the northeastern extremity of the Sea of Azov.

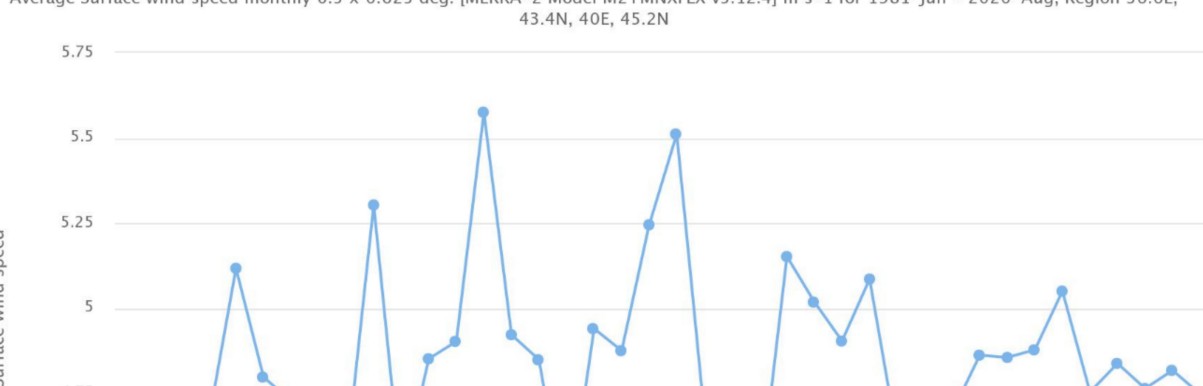

**Figure 13.** Interannual variability of the wind speed (m/s) in the northeastern Black Sea in summer from 1981–2020 based on the MERRA-2 M2TMNXFLX v5.12.4 Model.

The coastal zone is one of the most dynamic parts of the seas and oceans. It is here that the powerful wave energy transmitted to them by the wind is extinguished. The result of energy dissipation is the formation of strong currents and complex systems of intensive water exchange, the formation and movement of underwater bars, suspending and transfer of huge masses of sediments, and all these processes occur in a complex and in a variety of combinations. The state of the shores, especially beaches and coastal bottom relief, is largely determined by the nature of sediment movement in the coastal zone under the influence of wind waves, swells, and currents, therefore, in the context of the expansion of economic activity on the coast, scientific knowledge about the hydro- and litho-dynamic processes of the coastal zone and the transfer of huge masses of coastal sediments is extremely important [58,59].

The power and impact of winter and summer storms on the shores and beaches in Novorossiysk are shown in Figures 14 and 15a. In both cases, wind gusts were between 15 and 20 m/s. Strong winds in the summertime produce high waves along the shore which is dangerous for swimming, as well as water becomes turbid due to a high concentration of suspended matter even at a pebble beach. Winter storms, which usually are more severe, have also a negative impact on coastal tourism in summer because they destroy beaches (Figure 15a), which requires considerable funding for the reconstruction of beaches and infrastructure. One of the specific examples is the progressive destruction of the 1200 m long beach constructed in 2013 at the Olympic Park in Sochi before the opening of the Olympic Winter Games in 2014. During the past eight years, half of this beach was destroyed due to anthropogenic impacts (construction of a small port, which prevents natural movement of pebble and sediment from Mzymta Mouth along the coast) and the impact of storm surges. The authorities of the City of Sochi are forced to dump 25,000 tons of pebbles and rubble on this beach annually so that it does not erode.

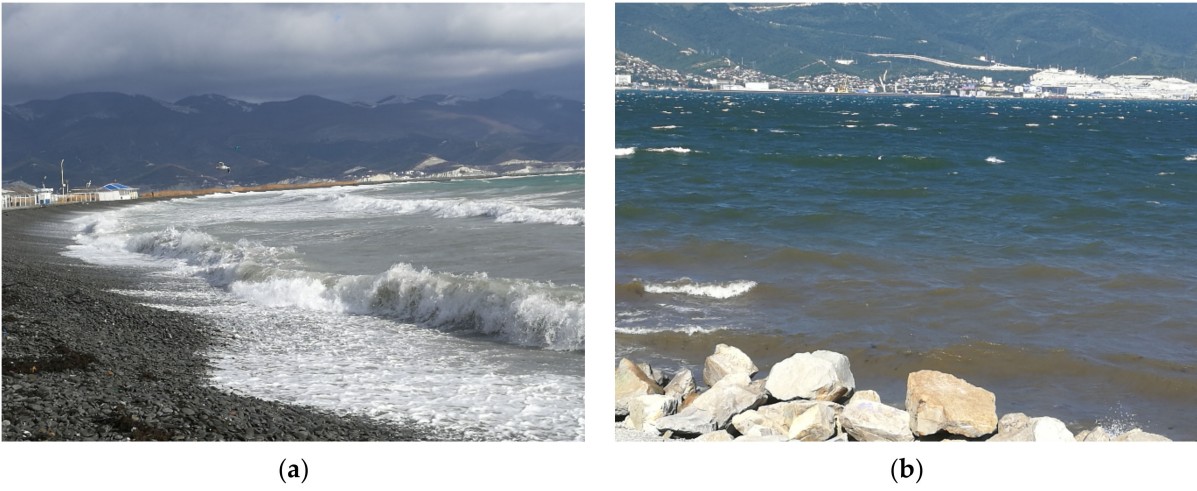

**Figure 14.** (**a**) Winter storm in Novorossiysk on 22 January 2021 caused by southerly winds; (**b**) summer storm in Novorossiysk on 9 July 2021 caused by northeasterly winds (Nord-Ost).

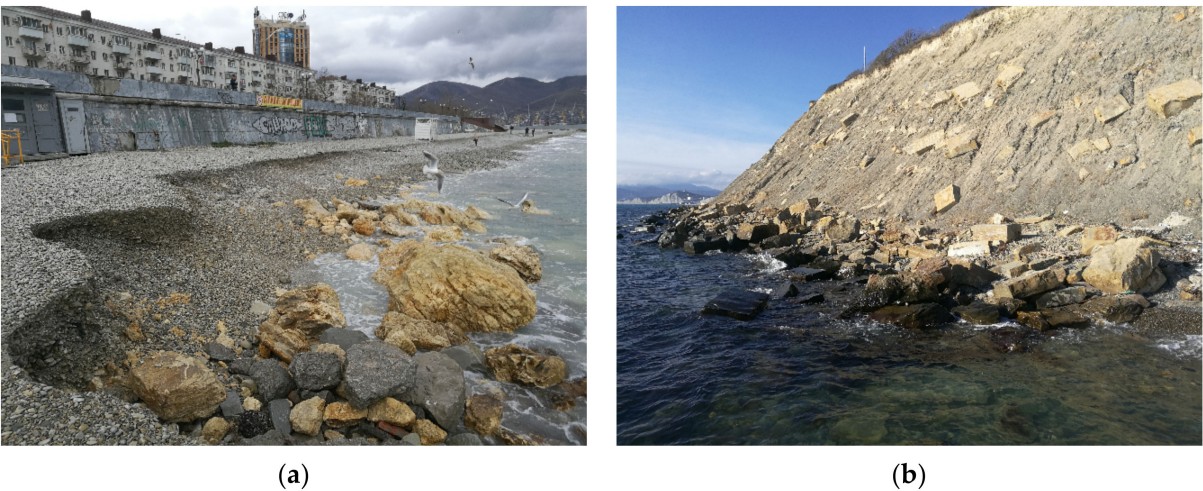

**Figure 15.** (**a**) Beach erosion in the City of Novorossiysk after a winter storm on 28 January 2021; (**b**) coast erosion in Shirokaya Balka near Novorossiysk due to landslide and talus processes, 4 February 2021.

Coastal and beach erosion is one of the negative processes that occur in the coastal zone due to wind, waves, and swells forcing and substantially impacting coastal infrastructure [54]. In any part of the coastal zone erosion can occur, since the resistance to erosion is not constant and this process depends on the ratio of the wave height, wave period, bottom slope and sediment structure, direction, and frequency of storm surges. Climatic changes in wind speed and direction not only near the coast (generation of wind waves) but also at a distance of tens and hundreds of kilometers (generation of swell) can affect the erosion of the coast even in those places where it was not previously observed [60].

For the Black Sea coast of Russia, the greatest influence on the formation of storm waves is exerted by the winds from the south, south-west, and west directions, and the prevalence of storm waves is from west, south-west, and south directions. Wave heights from these directions can reach 6.0 m and more with a maximum recorded 12.3 m. At such a wave height, the longest waves of 120–200 m are observed [14,61].

The ongoing degradation of the natural coastal zone and beaches on the Russian Black Sea coast is registered for several decades (Figure 15b). The tendencies of storm destruction of the coast may increase in connection with the ongoing rise in the level of the Black Sea, and the shortage of solid material in the coastal zone, which is caused by the fact that under intense hydrodynamic impact its entrainment exceeds its input [58,59,61]. For sandy

beaches northward of Anapa (from Anapa to Veselovka) 47 km long, it was shown that the destructive effects of wind waves and swell can be comparable, and the role of swell has been increasing over the past 40 years. As a result, from the mid-1960s to the 2010s, 20 to 80 m of beaches have already eroded [58,59]. Terrestrial runoff caused by heavy rains also leads to abrasion-landslide processes in the coastal zone (Figure 15b). In this relation, vulnerability maps for coast and beach erosion for the whole coastal zone of the Russian part of the Black Sea should be calculated on the basis of different scenarios of regional climate change causing changes in the wind–wave and precipitation regime.

The practical experience of coastal protection, as well as the results of hydro- and litho-dynamic studies, shows that the most effective way of coastal protection is the preservation and creation of wide beaches which play a role as smooth wave energy absorbers. It has been established that for the complete damping of wave energy of even a strong storm, which may occur once every 25 years, the width of the beach on the Black Sea coast should be on average 30–35 m. With a smaller width, the beaches are subject to erosion, and the coast will be destroyed [61].

Rip current is a very dangerous natural hydrodynamic phenomenon typical for some beaches with shallow water. It occurs when wind and breaking waves push surface water towards the beach, which causes a local slight rise in the water level along the shore. Then, this excess of seawater tends to flow back to the open sea perpendicular to the shore via the route of least resistance, such as slightly deeper parts or breaks in the bottom topography of a sand bar. Gravity initiates a rip current through this gap tens meters wide with a velocity up to 3 m/s. Swimmers caught by a fast rip current begin to panic, try to swim against the water flow back to the shore, and finally exhaust themselves and drown. Rip current is a horizontal current that does not pull people under the water. Very often, rip currents are referred to incorrect terms such as "rip tides" and "undertow". For instance, the undertow occurs everywhere beneath shore-approaching waves, whereas rip currents are localized narrow offshore currents occurring at certain locations along the beach. Rip currents have become a serious problem for the beaches from Anapa to Vityazevo, which kill several people yearly. Every year, these shallow sandy beaches, which were known for decades as the safest beaches for children, are periodically closed due to the sporadic occurrence of rip currents. Tragic news arrived when this paper was under preparation. On 5 July 2021, during a storm at Vityazevo beach near Anapa, three men drowned while trying to save a child who was being carried away by a rip current. Rip currents may intensify due to an increase in wave energy (both wind waves and swell) and a related change in the bottom topography near the shoreline.

*3.5. Coastal Upwelling*

Coastal upwelling is a natural hydrodynamic phenomenon, which is caused by favorable alongshore (the coast should be on the left in the Northern Hemisphere) and offshore winds. It leads to the upwelling of relatively cold waters from deep layers right in the coastal zone with a temperature minimum near the shore. This is a large-scale process, which in the coastal zones of North-West and South-West Africa, and North-West and South-West America are thousands of km long, a hundred km wide, and in some places are observed all year round with seasonal variability of its intensity at northern and southern limits of the upwelling zone. In the Black Sea, there are no permanent upwelling zones, upwelling can arise sporadically at all coastal zones when favorable winds start to blow [62–66]. The main problem for beach tourism concerns the fact that during the upwelling event, the SST may drop by 10–15 °C from about 25 °C, which is a typical and comfortable temperature at the coasts of the Black Sea, and last from several days to several weeks. As a result, people cannot enter water even at an air temperature of 30 °C. The most known analog of this problem is the western coast of Portugal, where during summer coastal upwelling occurs every year, and it is the main cause why, at these coasts of Portugal, there are no resort areas while at the same latitudes in Spain, along the

Mediterranean coast, there are dozens of well-known resort areas and beach tourism is very popular.

Silvestrova [65] reported that from 1979 to 2016 120 cases of coastal upwelling were registered near Gelendzhik, which is one of the most important resort areas on the Black Sea coastal zone of Russia. In 51 cases, the SST drop was greater than 5 °C. According to this research, strong upwelling near Gelendzhik occurs every year with no evident tendency to increase or decrease in the frequency of appearance. In 1987, eight upwelling events were recorded, in 1995, 1996, 2004, 2009, and 2013—5–6 events. Stanichnaya and Stanichny [66] investigated the occurrence of coastal upwelling in the summer season between 15 May and 15 September. They showed that the average time of one upwelling event is of 5–7 days, while in some cases it can last 14–15 days. For the coastal zone of Crimea, from 1997 to 2011, the number of upwelling events varies from 5 to 12 per year (summer), the number of strong upwelling events with an SST drop of more than 5 °C—from 1 to 5 events, and the total duration—from 20 (2002) to 70 days (2011). Minimal SST values reached 8–9 °C, and the maximum difference with the surrounding waters was 15–16 °C. For instance, only during summer 2003, coastal upwelling occurred on 22–23 May, 7–11 June, 13–20 June, 22–25 June, 27 June–2 July, 7–21 July, 22–25 July, 13–16 August, and the SST near the shoreline varied from 10 to 17.5 °C instead of the typical 20–24 °C in-between the upwelling events [66].

Two examples of these upwelling events that have occurred along the coast of the Crimea and the Krasnodar Krai right during the summer tourist season are shown in Figure 16. On 30 July 2017, a strong upwelling was observed between Sevastopol and Yalta, where the SST was as low as 11–12 °C, while in the surrounding waters it was as high as 24–25 °C (Figure 16a). On 2 July 2018, the upwelling was detected along the Caucasian coast from Abkhazia in the south to Novorossiysk in the north (Figure 16b). Then, from 2 to 10 July it propagated from south to north reaching the area between Anapa and the Kerch Strait. The SST along the shore was 20–21 °C, while in the offshore waters it was 27 °C.

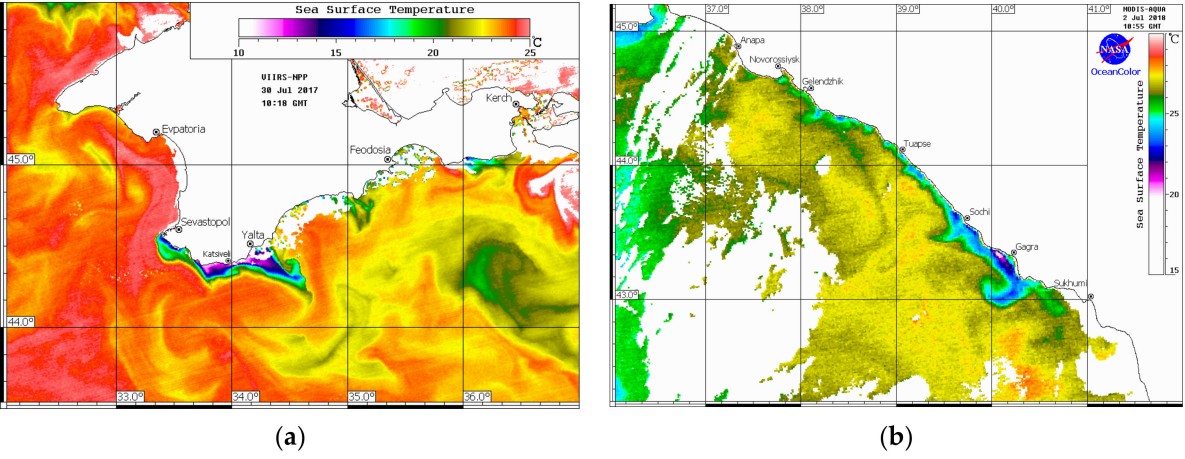

(**a**)  (**b**)

**Figure 16.** (**a**) Coastal upwelling (shown in SST) at the southern coast of the Crimea on 30 July 2017 (VIIRS-NPP, 10:18 GMT); (**b**) coastal upwelling (shown in SST) along the coast of the Krasnodar Krai on 2 July 2018 (MODIS-Aqua, 10:55 GMT) (courtesy by D.M. Soloviev, Marine Hydrophysical Institute).

*3.6. Sea Level Rise*

Today, sea-level rise is not a serious problem for most of the shores of the Black Sea, but a combined effect of the sea-level rise and wave impact on the shores and beaches have already led to the destruction of beaches and shore tourism infrastructure in many places of the Russian coastal zone in the Black Sea and the Sea of Azov [61].

Ginzburg et al. [35] analyzed interannual changes in the level anomalies of the Black and Azov Seas according to the data of the along-track altimetry measurements of Topex/Poseidon and Jason-1, Jason-2, and Jason-3 satellites in the period of 1993–2020. Long-term variability of the Black Sea level is characterized by alternating periods of its

rise and fall. At different time intervals, the rate of the level rise varied from the minimum value of +2.64 cm/year from January 1993 to June 1999 to the maximum +25.74 cm/year from August 2012 to July 2013; the rate of the level drop varied from −1.23 cm/year from June 1999 to April 2003 to −8.59 cm/year from June 2004 to February 2008. The average for the 1993–2020 period linear trend of the Black Sea level is of +0.32 ± 0.16 cm/year, which was found to be approximately 2.5 times lower than in 1993–2012 (+0.82 ± 0.18 cm/year), although 1.5 times more than from the 1920s to the mid-1990s (0.17–0.18 cm/year). For the northeastern part of the Black Sea, Lebedev et al. [40] showed that for the time interval of 1993–2015, the Black Sea level off the coast of the Krasnodar Krai grew at an average rate of 0.29 ± 0.03 cm/year, with an increase in rates from the Kerch Strait (0.28 cm/year) to Adler (0.31 cm/year). An average increase in the Black Sea level of 3.2 mm/year is consistent with the average world ocean sea-level rise [17].

### 3.7. Algal Bloom and Introduced Species

Algal bloom may be regarded as biological pollution of the coastal zone, which occurs as a result of an excess supply of nutrients (mainly nitrates and phosphates) in the coastal zone with river and land runoff, and high values of the SST. In the Black Sea, this problem occurs almost every year in the northwestern part of the Black Sea in the region of Odessa, and on shallow water sandy beaches of Anapa on the Russian coast [50,67]. The abundance of algae in the upper layers of the water makes the water green, not transparent, and entering into the water is unpleasant, and in some cases even dangerous (when swallowed), as it may contain toxic algae. In addition, the algae washed ashore starts to rot and the beach becomes unpleasant until cleaning (Figure 17).

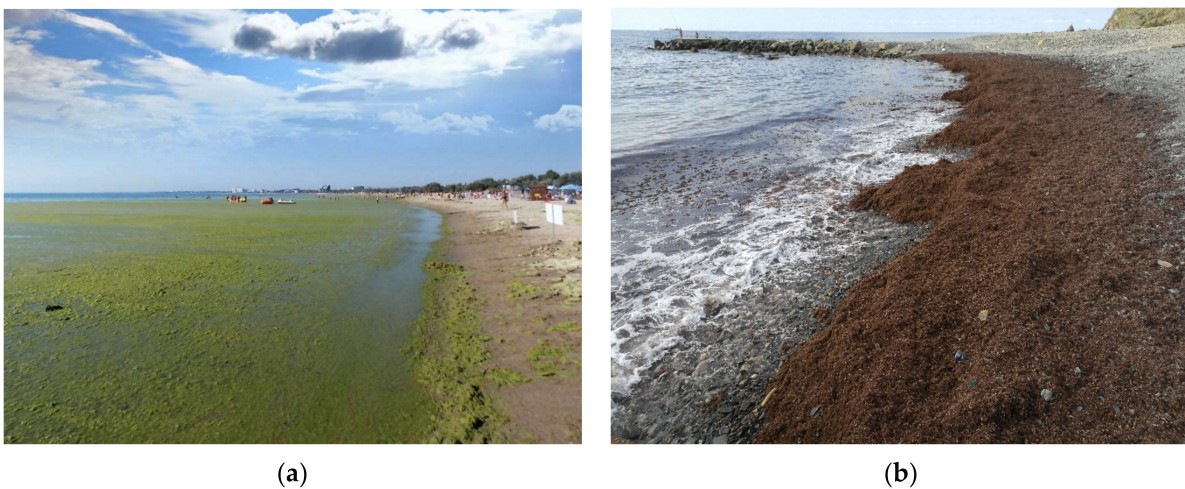

(**a**)          (**b**)

**Figure 17.** (**a**) Typical summer green algal bloom on beaches of Anapa (https://turvopros.com/kogda-cvetet-more-v-anape, accessed on 16 July 2021); (**b**) concentration of brown algae in water and dead algae washed ashore with a storm near Dyurso on 11 October 2020.

Introduced species is yet another problem for the Black Sea and for the development of coastal tourism in particular. Many alien species have become established in the Black Sea since the middle of the 20th century due to a discharge of ballast water from ships coming from adjacent seas and even oceans, as well as due to regional warming which facilitates northward expansion of species from the Mediterranean Sea. The most dramatic example of alien species introduction to the Black Sea was the invasion of a gelatinous predator, the polymorphic ctenophore *Mnemiopsis leidyi* (Figure 18a), and later on, the accidental invasion of ctenophore *Beroe ovata* (Figure 18b) [68]. On the international level, it is suggested that *Mnemiopsis* was brought to the Black Sea with ballast waters from the coastal regions of North America at the beginning of the 1980s. By 1988, *Mnemiposis* had spread over the entire Black Sea area and showed an enormous abundance outburst in

the fall of 1989 [68–71]. This led to a decrease in the biomass, abundance, and species diversity of edible zooplankton, fish larvae, and eggs, which are the principal food objects of *Mnemiopsis*. As a result, by 1991, the commercial fish catches had decreased 10 times; this especially referred to the anchovy, Mediterranean horse mackerel, sprat, as well as higher trophic levels—predator fishes and dolphins, who feed mainly on anchovies and sprats [68]. From the Black Sea, *Mnemiopsis* spread to the seas of Azov and Marmara, and in 1999—to the Caspian Sea. Then, in 1997, a new invader—ctenophore *Beroe ovata*—was first found in the northwestern part of the Black Sea, and in August 1999, the first outburst in the *Beroea* development over the entire Black Sea was observed. This predator feeds, first of all, on *Mnemiopsis*, thus, *Beroe* radically reduced the *Mnemiopsis* population and this resulted in the restoration of the Black Sea ecosystem within several years [68].

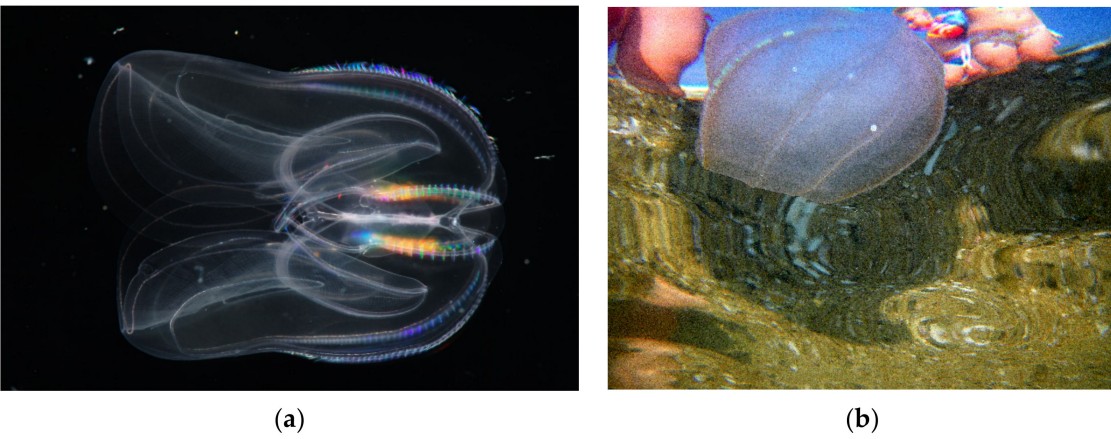

(**a**)           (**b**)

**Figure 18.** (**a**) *Mnemiopsis leidyi* (Courtesy by Bruno C. Vellutini—Own work, CC BY-SA 3.0, https://commons.wikimedia.org/w/index.php?curid=30155106, accessed on 18 July 2021); (**b**) *Beroe ovata* (Courtesy by Cristian Chirita—Own work, GFDL, https://commons.wikimedia.org/w/index.php?curid=10994369, accessed on 18 July 2021).

Both jellies are placed in the palm of your hand and do not pose any threat to adults or children, however, their number in different years reached such values when there were several dozens of them in one cubic meter of water. Swimming in such water is extremely unpleasant.

## 4. Discussion

Warming of the Black Sea coast of Russia, which includes both air temperature and sea surface temperature, is an evident positive factor that will favor the development of coastal tourism. The observed warming is accompanied by an expansion of the tourist season from June–August to September as well, which is already comparable with June by the air temperature. This will lead to a significant increase in water and electricity consumption related to an increase in the need for air conditioning and lighting. The lighting issue comes from the fact that time in the European part of Russia was established in such a way that, for example, in Novorossiysk on 22 June the sunrise is at 04:43 (local time) and sunset is at 20:18, and on 22 September, the sunrise is at 06:16 and sunset is at 18:26. It means that there is a lot of sun in the early morning and darkness in the early evening. Firstly, it is inconvenient for tourists who usually sleep in the morning, and secondly in the evening, when the air and water are still warm, swimming or walking in the dark is not very pleasant. This is an additional problem concerning the energy supply for lighting in the evening.

Regional climate change in the Black Sea is accompanied by the intensification of extreme weather events. The First [18] and Second [19] Roshydromet assessment reports on climate change and their consequences on the territory of the Russian Federation confirm these forecasts. Kostianoy et al. [38] showed that from 1950 to 2015, on average, heat waves in the Eastern Black Sea became a bit stronger (up to 4–6 °C hotter than the norm), their

frequency doubled for anomalous events exceeding 1 SD and reached 28–32 events per year, while strong events exceeding 2 SD increased 10 times from 1–2 to 12–14 per year. The average duration of extreme events with positive anomalies exceeding 1 SD increased from 2.5 to 3.5 days, and with anomalies exceeding 2 SD increased from 1 to 2 days.

These facts may result in health issues for the local population and tourists due to heatstroke from overheating from the sun, as well as in problems with water and energy supply. One such example is a draught on the coasts of the Black Sea, which occurred in all the Black Sea countries and lasted from summer 2019 to December 2020. The draught was caused by a lack of rainfall everywhere. The draught resulted in shallowing of rivers, freshwater reservoirs, and a significant decrease in the groundwater level around the Black Sea (Figure 19). For example, in Turkey, the water storage level in water reservoirs was the lowest during the past 15 years, and the year 2020 was the driest for the past 5 years [72]. As a result, during two tourist seasons in several coastal cities of the Krasnodar Krai and Crimea, there were restrictions in the water supply to housing when water was given to houses during 3 h in the morning and 3 h in the evening. The situation was aggravated by a large number of tourists in the summer of 2020 because of restrictions to travel to resorts of Turkey, Egypt, Greece, Cyprus, and other countries related to the COVID-19 pandemic.

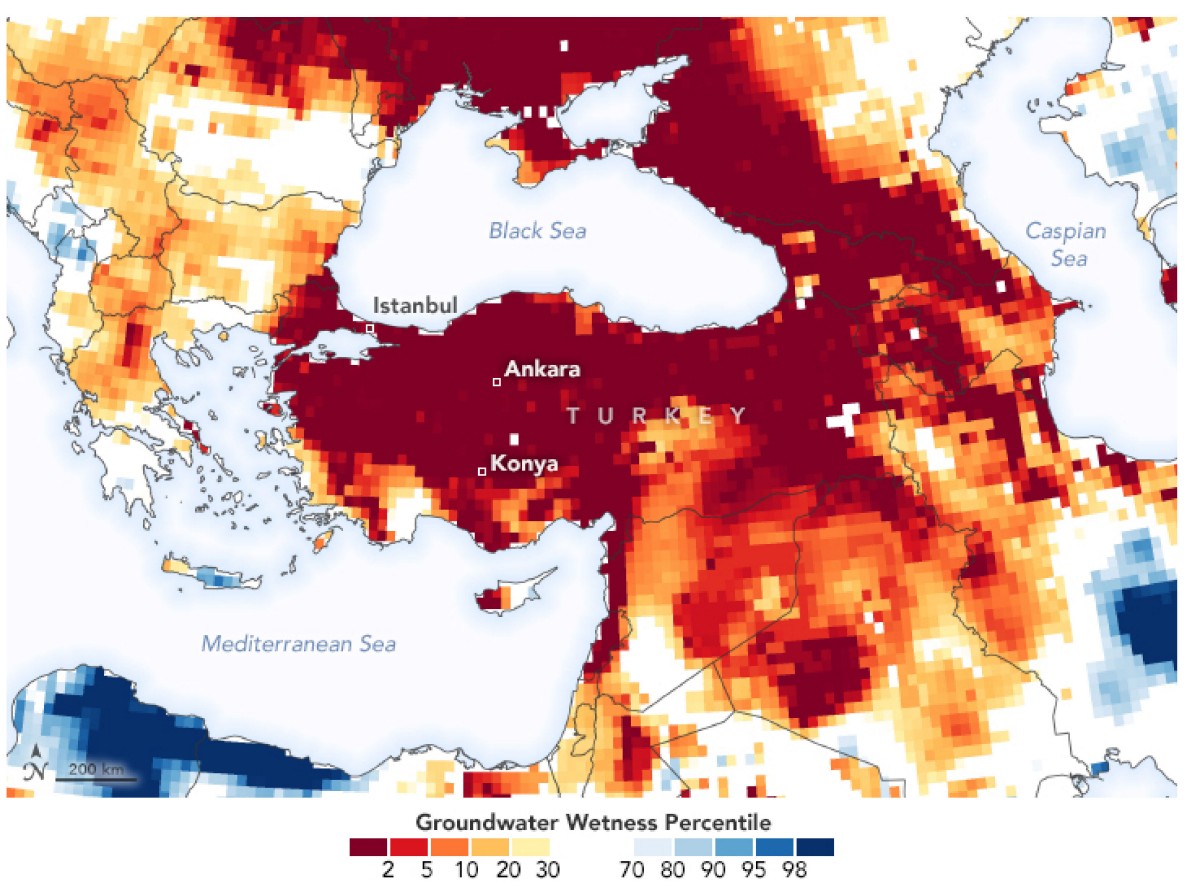

**Figure 19.** Shallow groundwater storage as of 11 January 2021 in the Black Sea Region and the Middle East, as measured by the Gravity Recovery and Climate Experiment Follow On (GRACE-FO) satellites. The color palette depicts the groundwater wetness percentile, i.e., how the amount of groundwater compares to long-term records (1948–2010). Blue areas have more water than usual, and orange and red areas have less groundwater. Credit: NASA's Earth Observatory [72].

Further increase in air temperature and intensity of drought periods will likely lead to more frequent forest and wildfires. This, in turn, poses a threat to the safety and security of the local population, will impact the attractiveness of the landscape, and the ultimate decision for tourists to choose this given location. This is already seen in such countries as Portugal, Spain, France, Italy, Montenegro, Greece, and Turkey [73].

We can also mention the recent paper by Klueva et al. [28] dealing with a forecast of the Tourism Climate Index (TCI), which merges tourism-related climate factors into a single index, for different regions of Russia, including the Black Sea coast of Russia. TCI is based on the air temperature, humidity, wind speed, atm precipitation, and time of sunshine. This is a very interesting study, which, for the coast of the Black Sea, shows that periods with TCI > 70 (which is characteristic for tourist comfort) will decrease from 90 days in 1981–2010 to about 30 days by 2050, and will tend to zero by the end of the 21st century. The main cause of this catastrophic decrease in TCI is a sharp rise in the expected air temperature which will be uncomfortable for people. Our research shows a set of additional factors like extreme weather events (heat waves, heavy rains, tornado), hydrological features like upwellings, river plumes, rip currents, wave impact, coastal erosion, biological factors, such as algal bloom, and introduced species. Both studies complement each other.

In recent years, torrential rains have become a real disaster for the resorts of the Krasnodar Krai and the Crimea, and although, on average, the amount of precipitation in summer has not increased over the past decade, the scale of the disaster caused by almost every rainstorm and flash flooding is enormous. We can recall heavy rains in the region on 17–18 June, 26 June, and 4–5 July 2021 that in some places were comparable by their disastrous impact to the heavy rain and flash flood in Western Germany, Southern Netherlands, and Southeastern Belgium on 15 July 2021. Even if it is one case for the entire summer holiday season, it can have disastrous consequences for the resort infrastructure, beaches, roads, and railways. For the coastal zone of the Krasnodar Krai, flooded railway stations and roads, damage to roads and railways bridges, as well as landslides and mudflows on these transport lines may cut all kinds of transport from several hours to several days because this is the only road and railways going along the coast from Dzhubga to Tuapse, and then to Sochi [74].

Change in the atmospheric circulation may lead to a change in cloudiness which can have a considerable effect on the attractiveness of a location for tourists. Such an example already exists on the Black Sea—the eastern part of the northern coastal zone of Turkey where the air temperature and sea surface temperature are even warmer than in the Krasnodar Krai, but which does not have sea resorts because this zone is almost always covered by clouds. This eastern part receives up to 2540 mm of rainfall per year, which makes it the wettest region in the country [75].

We did not find an increase in average wind speed during summer, but even at the present wind force, its direction can change in a way that will lead to serious erosion of the coasts and beaches due to the wave impact, to the appearance of coastal upwelling in the locations of the coastal zone of the Krasnodar Krai and the Crimea, where before it was not registered. This impact will be intensified with a progressive sea-level rise in the Black Sea. Upwelling may last for 1–2 weeks and the SST may drop by 10–15 °C. We consider upwelling as a serious threat to the development of coastal and beach tourism because there are many examples in the world where large-scale upwelling systems prevent its development even if the appropriate infrastructure exists. In Europe, the best example is the western coast of Portugal, and in North America—the coast of California. Today, the impact of algal bloom can be regarded as a local problem for the region of Anapa shallow water sandy beaches and along the coasts of the shallow Sea of Azov. The appearance of newly introduced species in the Black Sea is quite possible which is explained now by the warming of the sea and the establishment of favorable conditions for new species originated from the Mediterranean and Red Seas.

As we have shown, the frequency of extreme events is growing, so there is the variety of unpleasant events that make coastal tourism activities unpleasant or uncomfortable. Tourists might well fall into the situation where, if they visit the Black Sea coast of Russia every year, they will repeatedly catch one or another such event and will decide to go to another tourist destination the following year. Imagine that during one vacation, the duration of which for most vacationers is 1–2 weeks, tourists experience abnormal heat,

lack of fresh water, and problems with electricity, in another year—heavy rain and flash flood accompanied by large river plumes, in the third year—strong wind and waves, in the fourth year—swell from distant storms, in the fifth year—an upwelling event for several days or even weeks, in the sixth year—algal bloom of waters or the invasion of jellyfish, and so on. These events might as well happen all in one summer. The likelihood of ruining the only vacation of the year is growing. Some of the unlucky people will next time think about where to go on vacation—to the Crimea, the Caucasus, Turkey, Egypt, or other countries. Until now, the decision was mainly dependent on the financial capabilities of tourists and on the price/quality ratio of the hotel and tourist services provided. At present, the weather factor also interferes in the decision-making process, which is a direct consequence of regional climate change. It will have to be taken into account not only by vacationers but also by the authorities and the tourism industry.

## 5. Conclusions

In the paper, we did not investigate the impact of anthropogenic factors, geopolitical and socio-economic processes, and the COVID-19 pandemic that also plays an important role in the sustainable development of coastal tourism on the Black Sea coasts of Russia. For instance, according to World Travel and Tourism Council [3], in 2020, due to the consequences of the COVID-19 pandemic, the total GDP contribution dropped from 10.4% to 5.5% (by 49.1% from USD 9.17 to 4.67 trillion), 62 million jobs were lost, representing a drop of 18.5%, leaving just 272 million employed across the tourism sector globally, compared to 334 million in 2019. The Council says: "The threat of job losses persists as many jobs are currently supported by government retention schemes and reduced hours, which without a full recovery of Travel and Tourism could be lost. Domestic visitor spending decreased by 45%, while international visitor spending declined by an unprecedented 69.4%" [3].

In Russia, according to World Travel and Tourism Council [3], in 2020, the total contribution of travel and tourism to GDP dropped by 47% from USD 75.5 to 40.1 billion, 205.7 thousand people ($-5.1\%$) lost their jobs in the tourist sector, international visitor spending dropped by 69.6% from USD 14.8 to 4.5 billion, and domestic visitor spending dropped by 43.9% from USD 37.9 to 21.3 billion. The share of international visitors concerns the main tourist destinations in Russia—Moscow and St.-Petersburg, while the domestic visitor share reflects mainly coastal tourism in the Black Sea. New COVID-19-related restrictions for tourists on the coasts of the Black Sea for the summer season of 2021 will continue to considerably impact socio-economic conditions in the Krasnodar Krai and Crimea.

The above-mentioned example shows that external factors like geopolitical and socio-economic processes, global or domestic pandemics can suddenly cut one of the most important sectors of the national and global economy by around 50%. Regional climate change is one of these factors which affects the sustainable development of coastal tourism but is usually ignored. It would be wrong to believe that climate change is happening very slowly, so its effects can be felt in decades. The intensification of extreme weather conditions and negative natural processes in the atmosphere and the sea already today presents tourists with a serious choice—where is the best place to spend their next summer vacation? The answer to this question is not as obvious as it was in the previous decades.

The Russian Federation continues to invest significant funding in the development of infrastructure related to coastal tourism. On 2–5 June 2021, at the St.-Petersburg Economic Forum, a decision was taken to invest around USD 1.12 billion in the development of resorts and tourism in the Krasnodar Krai, in particular, in the construction of new resorts in Sochi and Anapa [76]. At the beginning of July 2021, the Russian Government announced a plan for the construction of a new tourist complex near Anapa called "New Anapa", which envisages the construction of 50 3–5 stars hotels, as well as restaurants, shops, roads, and other infrastructure which has to provide year-round tourist services. It is expected that it will provide 25,000 new jobs and will attract an additional 2–3 million tourists per year. This ambitious plan should be performed in three years and will cost

around USD 3.5 billion [77]. The Russian Ministry of Transport and Russian Railways are developing plans to construct a new railroad from Goryachy Klyuch to Sochi (which presently extends along the shoreline from Tuapse to Sochi) through mountains and a new automobile road from Dzhubga to Sochi (which also extends along the coast) several kilometers from the beach and urban infrastructure. The railway project will cost around USD 19 billion and will be completed by 2030, the automobile road project is estimated to be USD 32 billion and will be accomplished by 2035 [78]. As usual, all these plans did not take into account the negative factors that can arise in the near future from regional climate change. The comprehensive analysis of these factors would make it possible to make plans scientifically sound and investments much more effective.

We believe that this current research will serve as the first step in the multifactor complex study of this important matter. The present and future research are funded within the framework of the Russian Ministry of Science and Education Project "Comprehensive studies of the ecological state of waters of the coastal zone of the northeastern shelf of the Black Sea in the framework of participation in the international project DOORS" related to collaboration with the DOORS (Developing Optimal and Open Research Support system to unlock the potential for blue growth in the Black Sea) Project—a large international project started on 1 June 2021 within the framework of the Blue Growth thematic competition of the HORIZON-2020 Program of the European Union. The DOORS consortium includes 37 organizations from 17 countries, including the P.P. Shirshov Institute of Oceanology of the Russian Academy of Sciences. It is expected that based on the results of the project, specific practical recommendations will be prepared for optimization of maritime economic activity and the Blue Growth in all the Black Sea countries. We hope that our research related to climate change impacts on coastal tourism will be a base for such recommendations because coastal tourism is a major economic sector for all the Black Sea countries.

In this study, we focused on the assessment of climatic conditions for beach-based coastal tourism which makes it comfortable for tourists to spend time on the beach or in the water. This research has limitations as it has not covered many related aspects such as, for example: impact of climate change on wildlife in the region, detailed analysis of the impact of climate change on railway and road infrastructure, airports, and consequences for the touristic attractiveness of the region; the impact of climate change on yachting, cruising, boating, surfing, sup surfing, windsurfing, sailing, etc. This research also does not suggest any management strategies as this was out of the scope of our research. This paper has not either discussed the economic consequences of such climate change impacts on the region as this merits a separate study.

For future research, the authors are planning to develop the following directions of research: (1) detailed investigation of individual factors and their interannual variability; (2) investigation of local peculiarities for different parts of the Black Sea coast of Russia; (3) collaboration with other Black Sea countries in the framework of the DOORS Project to investigate the impact of climate change on their coastal zone; (4) elaboration of recommendations for local authorities and tourism business to overcome potential negative consequences of regional climate change.

**Author Contributions:** Conceptualization, E.A.K. and A.G.K.; methodology, A.G.K.; formal analysis, E.A.K. and A.G.K.; investigation, E.A.K. and A.G.K.; resources, A.G.K.; data curation, A.G.K.; writing—original draft preparation, E.A.K.; writing—review and editing, A.G.K.; visualization, E.A.K.; supervision, A.G.K.; project administration, A.G.K.; funding acquisition, A.G.K. All authors have read and agreed to the published version of the manuscript.

**Funding:** This research was funded in the framework of the Russian Ministry of Science and Education Project "Comprehensive studies of the ecological state of waters of the coastal zone of the northeastern shelf of the Black Sea in the framework of participation in the international project DOORS" performed under the Agreement on the provision of a grant from the federal budget in the form of a subsidy 13.2251.21.0008. The APC was covered by the MDPI.

**Institutional Review Board Statement:** Not applicable.

**Informed Consent Statement:** Not applicable.

**Data Availability Statement:** All data used in the present analysis are freely available from the databases and websites mentioned in Materials and Methods Section.

**Acknowledgments:** We acknowledge collaboration with the DOORS Project (Grant Agreement ID 101000518) performed in the framework of the EU HORIZON-2020 Programme. We acknowledge the use of imagery from the Worldview Snapshots application (https://wvs.earthdata.nasa.gov), part of the Earth Observing System Data and Information System (EOSDIS). We are thankful to Dmitry M. Soloviev from Marine Hydrophysical Institute, who provided us with thermal imagery presenting recent upwelling events.

**Conflicts of Interest:** The authors declare no conflict of interest. The funders had no role in the design of the study; in the collection, analyses, or interpretation of data; in the writing of the manuscript, or in the decision to publish the results.

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
