# Peer review of "Regional Climate Change Impact on Coastal Tourism: A Case Study for the Black Sea Coast of Russia"

_hydrology, doi:10.3390/hydrology8030133_

Round 1
Reviewer 1 Report
This study reviews climate change impact on the Black Sea coastal tourism by assessing several hydrological, meteorological, and biological factors. This topic is a cross-disciplinary issue and deserves more attentions. Overall this study does cover more aspects of physical factors and less social economic factors in coastal tourism. Following are my recommendations to improve the quality of the paper.
- At the beginning of the manuscript, more references related to the vulnerability of coastal tourism are needed to be comparable to the later discussions of impact factors.
- In section 3.2, I think it is also important that the authors can also explain the relationship between air temperature and SST.
- In section 3.3, the arguments of heavy rain causing severe problems due to regional climate change is not persuasive. For example, how often do heavy rains cause serious problems recently compared to similar events earlier? Also do the scales of damages in recent years become larger than before? The trend of precipitation (Fig. 9) cannot support the argument of climate change in the study area.
- In section 3.4, by showing some of photos (Fig. 13 and 14) cannot really approve the impact of storms, wind, wave and swell in the study area. The authors may need to show inter-annual aerial photos of coastal zones and delineate the retreatment of coastal lines.
- In section 3.5, I think it is also important to describe how SST can affect upwelling.
- In section 3.6, sea level rise is sure a strong indicator related to climate change. How seriously does this phenomenon affect the coastal tourism in the study area?
- In section 3.7, as the authors state that algal bloom occurs almost every year. Then is this problem still related to climate change?
- The 2nd paragraph in the discussion is the rationale of the study which should be placed in the beginning of paper.
- The last paragraph in the discussion (line 747-750), why do the authors divide the impact factors into different years?
Reviewer 2 Report
Thank you for inviting me to review the manuscript: Regional climate change on costal tourism: A case study for the Black Sea Coast of Russia. The aim of the paper was to investigate different negative impacts of natural factors on costal tourism in the Black Sea Coast of Russia area.
The paper in interesting in the context of sustainable tourism development in the research area.
Even if the paper is focusing more on the natural factors impact on tourism development, I was expecting to see some data related to tourism supply and demand in the research area.
At the same time is not clear which is the novelty of the paper and what gap fulfills in the literature. Which are the managerial implications of the results? What are future research directions?
Best regards!
Reviewer 3 Report
- The research just reviews a series of potential negative environmental factors impact coastal tourism, but it doesn’t explain clearly how to evaluate the impact on the coastal tourism caused by climate change. There is lack of an overall assessment method to evaluate the impacts.
- The type of coastal tourism is diverse, include the full range of tourism, leisure, and recreational oriented activities and the infrastructure supporting coastal development. The research does not illustrate that what the impacts caused by climate change. Suggestion to explain which types of marine tourism impacted by climate change.
- It is recommended to increase more references about this issue in this paper.
- This study does not provide some management strategies.
Round 2
Reviewer 1 Report
The authors have responded well most of my comments.
Reviewer 3 Report
no comments